# Data-Efficient Augmentation for Training Neural Networks

**Tian Yu Liu**
Department of Computer Science
University of California, Los Angeles
tianyu@cs.ucla.edu

**Baharan Mirzasoleiman**
Department of Computer Science
University of California, Los Angeles
baharan@cs.ucla.edu

## Abstract

Data augmentation is essential to achieve state-of-the-art performance in many deep learning applications. However, the most effective augmentation techniques become computationally prohibitive for even medium-sized datasets. To address this, we propose a rigorous technique to select subsets of data points that when augmented, closely capture the training dynamics of full data augmentation. We first show that data augmentation, modeled as additive perturbations, improves learning and generalization by relatively enlarging and perturbing the smaller singular values of the network Jacobian, while preserving its prominent directions. This prevents overfitting and enhances learning the harder to learn information. Then, we propose a framework to iteratively extract small subsets of training data that when augmented, closely capture the alignment of the fully augmented Jacobian with labels/residuals. We prove that stochastic gradient descent applied to the augmented subsets found by our approach has similar training dynamics to that of fully augmented data. Our experiments demonstrate that our method achieves 6.3x speedup on CIFAR10 and 2.2x speedup on SVHN, and outperforms the baselines by up to 10% across various subset sizes. Similarly, on TinyImageNet and ImageNet, our method beats the baselines by up to 8%, while achieving up to 3.3x speedup across various subset sizes. Finally, training on and augmenting 50% subsets using our method on a version of CIFAR10 corrupted with label noise even outperforms using the full dataset. [1]

## 1   Introduction

Standard (weak) data augmentation transforms the training examples with e.g. rotations or crops for images, and trains on the transformed examples *in place of* the original training data. While weak augmentation is effective and computationally inexpensive, *strong* data augmentation (in addition to weak augmentation) is a key component in achieving nearly all state-of-the-art results in deep learning applications [31]. However, strong data augmentation techniques often increase the training time by orders of magnitude. First, they often have a very expensive pipeline to find or generate more complex transformations that best improves generalization [4, 13, 19, 36]. Second, *appending transformed examples* to the training data is often much more effective than training on the (strongly or weakly) transformed examples *in-place* of the original data. For example, appending *one* transformed example to the training data is often much more effective than training on *two* transformed examples *in place* of every original training data, while both strategies have the same computational cost (*c.f.* Appendix D.6). Hence, to obtain the state-of-the-art performance, multiple augmented examples are added for every single data point and to each training iteration [12, 36]. In this case, even if producing transformations are cheap, such methods increases the size of the training data by orders of magnitude.

---

[1]Our code can be found at https://github.com/tianyu139/data-efficient-augmentation

36th Conference on Neural Information Processing Systems (NeurIPS 2022).

As a result, state-of-the-art data augmentation techniques become computationally prohibitive for even medium-sized real-world problems. For example, the state-of-the-art augmentation of [36], which appends every example with its highest-loss transformations, increases the training time of ResNet20 on CIFAR10 by 13x on an Nvidia A40 GPU (*c.f.* Sec. 6).

To make state-of-the-art data augmentation more efficient and scalable, an effective approach is to carefully select a small subset of the training data such that augmenting only the subset provides similar training dynamics to that of full data augmentation. If such a subset can be quickly found, it would directly lead to a significant reduction in storage and training costs. First, while standard in-place augmentation can be applied to the entire data, the strong and *expensive transformations* can be only produced for the examples in the subset. Besides, only the transformed elements of the subset can be *appended* to the training data. Finally, when the data is larger than the training budget, one can train on random subsets (with standard in-place augmentation) and augment coresets (by strong augmentation and/or appending transformations) to achieve a superior performance.

Despite the efficiency and scalability that it can provide, this direction has remained largely unexplored. Existing studies are limited to fully training a network and subsampling data points based on their loss or influence, for augmentation in subsequent training runs [17]. However, this method is prohibitive for large datasets, provides a marginal improvement over augmenting random subsets, and does not provide any theoretical guarantee for the performance of the network trained on the augmented subsets. Besides, when the data contains mislabeled examples, augmentation methods that select examples with maximum loss, and append their transformed versions to the data, degrade the performance by selecting and appending several noisy labels.

A major challenge in finding the most effective data points for augmentation is to theoretically understand how data augmentation affects the optimization and generalization of neural networks. Existing theoretical results are mainly limited to simple linear classifiers and analyze data augmentation as enlarging the span of the training data [36], providing a regularization effect [3, 8, 33, 36], enlarging the margin of a linear classifier [28], or having a variance reduction effect [5]. However, such tools do not provide insights on the effect of data augmentation on training deep neural networks.

Here, we study the effect of label invariant data augmentation on training dynamics of overparameterized neural networks. Theoretically, we model data augmentation by *bounded additive perturbations* [28], and analyze its effect on neural network Jacobian matrix containing all its first-order partial derivatives [1]. We show that label invariant additive data augmentation *proportionally* enlarges but more importantly *perturbs* the singular values of the Jacobian, particularly the smaller ones, while maintaining prominent directions of the Jacobian. In doing so, data augmentation *regularizes* training by adding bounded but varying perturbations to the gradients. In addition, it *speeds up* learning harder to learn information. Thus, it prevents overfitting and improves generalization. Empirically, we show that the same effect can be observed for various strong augmentations, e.g., AutoAugment [6], CutOut [9], and AugMix [12].[2]

Next, we develop a rigorous method to iteratively find small weighted subsets (coresets) that when augmented, closely capture the alignment between the Jacobian of the full augmented data with the label/residual vector. We show that the most effective subsets for data augmentation are the set of examples that when data is mapped to the gradient space, have the most centrally located gradients. This problem can be formulated as maximizing a submodular function. The subsets can be efficiently extracted using a fast greedy algorithm which operates on small dimensional gradient proxies, with only a small additional cost. We prove that augmenting the coresets guarantees similar training dynamics to that of full data augmentation. We also show that augmenting our coresets achieve a superior accuracy in presence of noisy labeled examples.

We demonstrate the effectiveness of our approach applied to CIFAR10 (ResNet20, WideResNet-28-10), CIFAR10-IB (ResNet32), SVHN (ResNet32), noisy-CIFAR10 (ResNet20), Caltech256 (ResNet18, ResNet50), TinyImageNet (ResNet50), and ImageNet (ResNet50) compared to random and max-loss baselines [17]. We show the effectiveness of our approach (in presence of standard augmentation) in the following cases:

- **When producing augmentations is expensive and/or they are appended to the training data:**

---

[2]We note that our results are in line with that of [30], that in parallel to our work, analyzed the effect of linear transformations on a two-layer convolutional network, and showed that it can make the hard to learn features more likely to be captured during training.

We show that for the state-of-the-art augmentation method of [36] applied to CIFAR10/ResNet20 it is 3.43x faster to train on the whole dataset and only augment our coresets of size 30%, compared to training and augmenting the whole dataset. At the same time, we achieve 75% of the accuracy improvement of training on and augmenting the full data with the method of [36], outperforming both max-loss and random baselines by up to 10%.

- **When data is larger than the training budget:** We show that we can achieve 71.99% test accuracy on ResNet50/ImageNet when training on and augmenting only 30% subsets for 90 epochs. Compared to AutoAugment [6], despite using only 30% subsets, we achieve 92.8% of the original reported accuracy while boasting 5x speedup in the training time. Similarly, on Caltech256/ResNet18, training on and augmenting 10% coresets with AutoAugment yields 65.4% accuracy, improving over random 10% subsets by 5.8% and over only weak augmentation by 17.4%.

- **When data contains mislabeled examples:** We show that training on and strongly augmenting 50% subsets using our method on CIFAR10 with 50% noisy labels achieves 76.20% test accuracy. Notably, this yields a superior performance to training on and strongly augmenting the full data.

## 2    Additional Related Work

Strong data augmentation methods achieve state-of-the-art performance by finding the set of transformations for every example that best improves the performance. Methods like AutoAugment [6], RandAugment [7], and Faster RandAugment [7] search over a (possibly large) space of transformations to find sequences of transformations that best improve generalization [6, 7, 21, 36]. Other techniques involve a very expensive pipeline for generating the transformations. For example, some use Generative Adversarial Networks to directly learn new transformations [2, 21, 24, 29]. Strong augmentations like Smart Augmentation [19], Neural Style Transfer-based [13], and GAN-based augmentations [4] require an expensive forward pass through a deep network for input transformations. For example, [13] increases training time by 2.8x for training ResNet18 on Caltech256. Similarly, [36] generates multiple augmentations for each training example, and selects the ones with the highest loss.

Strong data augmentation methods either replace the original example by its transformed version, or append the generated transformations to the training data. Crucially, appending the training data with transformations is much more effective in improving the generalization performance. Hence, the most effective data augmentation methods such as that of [36] and AugMix [12] append the transformed examples to the training data. In Appendix D.6, we show that even for cheaper strong augmentation methods such as AutoAugment [6], while replacing the original training examples with transformations may decrease the performance, appending the augmentations significantly improves the performance. Appending the training data with augmentations, however, increase the training time by orders of magnitude. For example, AugMix [12] that outperforms AutoAugment increases the training time by at least 3x by appending extra augmented examples, and [36] increases training time by 13x due to appending and forwarding additional augmented examples through the model.

## 3    Problem Formulation

We begin by formally describing the problem of learning from augmented data. Consider a dataset $\mathcal{D}_{train} = (\boldsymbol{X}_{train}, \boldsymbol{y}_{train})$, where $\boldsymbol{X}_{train} = (\boldsymbol{x}_1, \cdots, \boldsymbol{x}_n) \in \mathbb{R}^{d \times n}$ is the set of $n$ normalized data points $\boldsymbol{x}_i \in [0, 1]^d$, from the index set $V$, and $\boldsymbol{y}_{train} = (y_1, \cdots, y_n) \in \{y \in \{\nu_1, \nu_2, \cdots, \nu_C\}\}$ with $\{\nu_j\}_{j=1}^C \in [0, 1]$.

**The additive perturbation model.** Following [28] we model data augmentation as an arbitrary bounded additive perturbation $\boldsymbol{\epsilon}$, with $\|\boldsymbol{\epsilon}\| \leq \epsilon_0$. For a given $\epsilon_0$ and the set of all possible transformations $\mathcal{A}$, we study the transformations selected from $\mathcal{S} \subseteq \mathcal{A}$ satisfying

$$\mathcal{S} = \{T_i \in \mathcal{A} \mid \|T_i(\boldsymbol{x}) - \boldsymbol{x}\| \leq \epsilon_0 \ \forall \boldsymbol{x} \in \boldsymbol{X}^{train}\}. \tag{1}$$

While the additive perturbation model cannot represent all augmentations, most real-world augmentations are bounded to preserve the regularities of natural images (e.g. AutoAugment [6] finds that a 6 degree rotation is optimal for CIFAR10). Thus, under local smoothness of images, additive perturbation can model bounded transformations such as small rotations, crops, shearing, and pixel-wise transformations like sharpening, blurring, color distortions, structured adversarial perturbation [21]. As such, we see the effects of additive augmentation on the singular spectrum holds even under

real-world augmentation settings (*c.f.* Fig. 3 in the Appendix). However, this model is indeed limited when applied to augmentations that cannot be reduced to perturbations, such as horizontal/vertical flips and large translations. We extend our theoretical analysis to augmentations modeled as arbitrary linear transforms (e.g. as mentioned, horizontal flips) in Appendix B.5.

The set of augmentations at iteration $t$ generating $r$ augmented examples per data point can be specified, with abuse of notation, as $\mathcal{D}_{aug}^t = \{\bigcup_{i=1}^r (T_i^t(\boldsymbol{X}_{train}), \boldsymbol{y}_{train})\}$, where $|\mathcal{D}_{aug}^t| = rn$ and $T_i^t(\boldsymbol{X}_{train})$ transforms all the training data points with the set of transformations $T_i^t \subset \mathcal{S}$ at iteration $t$. We denote $\boldsymbol{X}_{aug}^t = \{\bigcup_{i=1}^r T_i^t(\boldsymbol{X}_{train})\}$ and $\boldsymbol{y}_{aug}^t = \{\bigcup_{i=1}^r \boldsymbol{y}_{train}\}$.

**Training on the augmented data.** Let $f(\boldsymbol{W}, \boldsymbol{x})$ be an arbitrary neural network with $m$ vectorized (trainable) parameters $\boldsymbol{W} \in \mathbb{R}^m$. We assume that the network is trained using (stochastic) gradient descent with learning rate $\eta$ to minimize the squared loss $\mathcal{L}$ over the original and augmented training examples $\mathcal{D}^t = \{\mathcal{D}_{train} \cup \mathcal{D}_{aug}^t\}$ with associated index set $V^t$, at every iteration $t$. I.e.,

$$\mathcal{L}(\boldsymbol{W}^t, \boldsymbol{X}) := \frac{1}{2}\sum_{i \in V^t}\mathcal{L}_i(\boldsymbol{W}^t, \boldsymbol{x}_i) := \frac{1}{2}\sum_{(\boldsymbol{x}_i, y_i) \in \mathcal{D}^t} \|f(\boldsymbol{W}^t, \boldsymbol{x}_i) - y_i\|_2^2. \tag{2}$$

The gradient update at iteration $t$ is given by

$$\boldsymbol{W}^{t+1} = \boldsymbol{W}^t - \eta\nabla\mathcal{L}(\boldsymbol{W}^t, \boldsymbol{X}), \quad \text{s.t.} \quad \nabla\mathcal{L}(\boldsymbol{W}^t, \boldsymbol{X}) = \mathcal{J}^T(\boldsymbol{W}^t, \boldsymbol{X})(f(\boldsymbol{W}^t, \boldsymbol{X}) - \boldsymbol{y}), \tag{3}$$

where $\boldsymbol{X}^t = \{\boldsymbol{X}_{train} \cup \boldsymbol{X}_{aug}^t\}$ and $\boldsymbol{y}^t = \{\boldsymbol{y}_{train} \cup \boldsymbol{y}_{aug}^t\}$ are the set of original and augmented examples and their labels, $\mathcal{J}(\boldsymbol{W}, \boldsymbol{X}) \in \mathbb{R}^{n \times m}$ is the Jacobian matrix associated with $f$, and $\boldsymbol{r}^t = f(\boldsymbol{W}^t, \boldsymbol{X}) - \boldsymbol{y}$ is the residual.

We further assume that $\mathcal{J}$ is smooth with Lipschitz constant $L$. I.e., $\|\mathcal{J}(\boldsymbol{W}, \boldsymbol{x}_i) - \mathcal{J}(\boldsymbol{W}, \boldsymbol{x}_j)\| \le L\|\boldsymbol{x}_i - \boldsymbol{x}_j\| \ \forall \boldsymbol{x}_i, \boldsymbol{x}_j \in \boldsymbol{X}$. Thus, for any transformation $T_j \in \mathcal{S}$, we have $\|\mathcal{J}(\boldsymbol{W}, \boldsymbol{x}_i) - \mathcal{J}(\boldsymbol{W}, T_j(\boldsymbol{x}_i))\| \le L\epsilon_0$. Finally, denoting $\mathcal{J} = \mathcal{J}(\boldsymbol{W}, \boldsymbol{X}_{train})$ and $\tilde{\mathcal{J}} = \mathcal{J}(\boldsymbol{W}, T_j(\boldsymbol{X}_{train}))$, we get $\tilde{\mathcal{J}} = \mathcal{J} + \boldsymbol{E}$, where $\boldsymbol{E}$ is the perturbation matrix with $\|\boldsymbol{E}\|_2 \le \|\boldsymbol{E}\|_F \le \sqrt{n}L\epsilon_0$.

# 4 Data Augmentation Improves Learning

In this section, we analyze the effect of data augmentation on training dynamics of neural networks, and show that data augmentation can provably prevent overfitting. To do so, we leverage the recent results that characterize the training dynamics based on properties of neural network Jacobian and the corresponding Neural Tangent Kernel (NTK) [14] defined as $\boldsymbol{\Theta} = \mathcal{J}(\boldsymbol{W}, \boldsymbol{X})\mathcal{J}(\boldsymbol{W}, \boldsymbol{X})^T$. Formally:

$$\boldsymbol{r}^t = \sum_{i=1}^n (1 - \eta\lambda_i)(\boldsymbol{u}_i\boldsymbol{u}_i^T)\boldsymbol{r}^{t-1} = \sum_{i=1}^n (1 - \eta\lambda_i)^t(\boldsymbol{u}_i\boldsymbol{u}_i^T)\boldsymbol{r}^0, \tag{4}$$

where $\boldsymbol{\Theta} = \boldsymbol{U}\boldsymbol{\Lambda}\boldsymbol{U}^T = \sum_{i=1}\lambda_i\boldsymbol{u}_i\boldsymbol{u}_i^T$ is the eigendecomposition of the NTK [1]. Although the constant NTK assumption holds only in the infinite width limit, [18] found close empirical agreement between the NTK dynamics and the true dynamics for wide but practical networks, such as wide ResNet architectures [37]. Eq. (4) shows that the training dynamics depend on the alignment of the NTK with the residual vector at every iteration $t$. Next, we prove that for small perturbations $\epsilon_0$, data augmentation prevents overfitting and improves generalization by proportionally enlarging and perturbing smaller eigenvalues of the NTK relatively more, while preserving its prominent directions.

## 4.1 Effect of Augmentation on Eigenvalues of the NTK

We first investigate the effect of data augmentation on the singular values of the Jacobian, and use this result to bound the change in the eigenvalues of the NTK. To characterize the effect of data augmentation on singular values of the perturbed Jacobian $\tilde{\mathcal{J}}$, we rely on Weyl's theorem [35] stating that under bounded perturbations $\boldsymbol{E}$, no singular value can move more than the norm of the perturbations. Formally, $|\tilde{\sigma}_i - \sigma_i| \le \|\boldsymbol{E}\|_2$, where $\tilde{\sigma}_i$ and $\sigma_i$ are the singular values of the perturbed and original Jacobian respectively. Crucially, data augmentation affects larger and smaller singular values differently. Let $\boldsymbol{P}$ be orthogonal projection onto the column space of $\mathcal{J}^T$, and $\boldsymbol{P}_\perp = \boldsymbol{I} - \boldsymbol{P}$ be the projection onto its orthogonal complement subspace. Then, the singular values of the perturbed

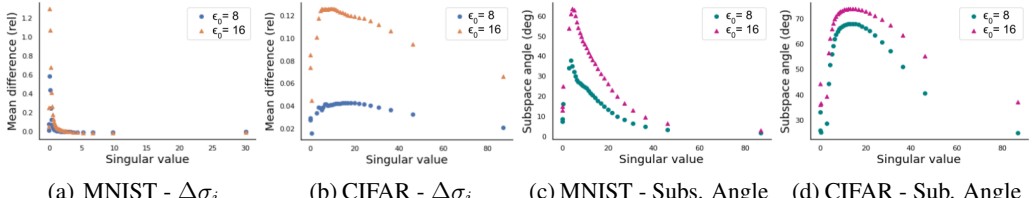

(a) MNIST - $\Delta\sigma_i$    (b) CIFAR - $\Delta\sigma_i$    (c) MNIST - Subs. Angle    (d) CIFAR - Sub. Angle

Figure 1: Effect of augmentations on the singular spectrum of the network Jacobian of ResNet20 trained on CIFAR10, and a MLP on MNIST, trained till epoch 15. (a), (b) Difference in singular values and (c), (d) singular subspace angles between the original and augmented data with bounded perturbations with $\epsilon_0 = 8$ and $\epsilon_0 = 16$ for different ranges of singular values. Note that augmentations with larger bound $\epsilon_0$ results in larger perturbations to the singular spectrum.

Jacobian $\tilde{\mathcal{J}}^T$ are $\tilde{\sigma}_i^2 = (\sigma_i + \mu_i)^2 + \zeta_i^2$, where $|\mu_i| \leq \|\boldsymbol{P}\boldsymbol{E}\|_2$, and $\sigma_{\min}(\boldsymbol{P}_\perp \boldsymbol{E}) \leq \zeta_i \leq \|\boldsymbol{P}_\perp \boldsymbol{E}\|_2$, $\sigma_{\min}$ the smallest singular value of $\mathcal{J}^T$ [32]. Since the eigenvalues of the projection matrix $\boldsymbol{P}$ are either 0 or 1, as the number of dimensions $m$ grows, for bounded perturbations we get that on average $\mu_i^2 = \mathcal{O}(1)$ and $\zeta_i^2 = \mathcal{O}(m)$. Thus, the second term dominates and increase of small singular values under perturbation is proportional to $\sqrt{m}$. However, for larger singular values, first term dominates and hence $\tilde{\sigma}_i - \sigma_i \cong \mu_i$. Thus in general, small singular values can become proportionally larger, while larger singular values remain relatively unchanged. The following Lemma characterizes the *expected* change to the eignvalues of the NTK.

**Lemma 4.1.** *Data augmentation as additive perturbations bounded by small $\epsilon_0$ results in the following expected change to the eigenvalues of the NTK:*

$$\mathbb{E}[\tilde{\lambda}_i] = \mathbb{E}[\tilde{\sigma}_i^2] = \sigma_i^2 + \sigma_i(1 - 2p_i)\|\boldsymbol{E}\| + \|\boldsymbol{E}\|^2/3 \tag{5}$$

*where $p_i := \mathbb{P}(\tilde{\sigma}_i - \sigma_i < 0)$ is the probability that $\sigma_i$ decreases as a result of data augmentation, and is smaller for smaller singular values.*

The proof can be found in Appendix A.1.

Next, we discuss the effect of data augmentation on singular vectors of the Jacobian and show that it mainly affects the non-prominent directions of the Jacobian spectrum, but to a smaller extent compared to the singular values.

## 4.2 Effect of Augmentation on Eigenvectors of the NTK

Here, we focus on characterizing the effect of data augmentation on the eigenspace of the NTK. Let the singular subspace decomposition of the Jacobian be $\mathcal{J} = \boldsymbol{U}\boldsymbol{\Sigma}\boldsymbol{V}^T$. Then for the NTK, we have $\boldsymbol{\Theta} = \mathcal{J}\mathcal{J}^T = \boldsymbol{U}\boldsymbol{\Sigma}\boldsymbol{V}^T\boldsymbol{V}\boldsymbol{\Sigma}\boldsymbol{U}^T = \boldsymbol{U}\boldsymbol{\Sigma}^2\boldsymbol{U}^T$ (since $\boldsymbol{V}^T\boldsymbol{V} = \boldsymbol{I}$). Hence, the perturbation of the eigenspace of the NTK is the same as perturbation of the left singular subspace of the Jacobian $\mathcal{J}$. Suppose $\sigma_i$ are singular values of the Jacobian. Let the perturbed Jacobian be $\tilde{\mathcal{J}} = \mathcal{J} + \boldsymbol{E}$, and denote the eigengap $\gamma_0 = \min\{\sigma_i - \sigma_{i+1} : i = 1, \cdots, r\}$ where $\sigma_{r+1} := 0$. Assuming $\gamma_0 \geq 2\|\boldsymbol{E}\|_2$, a combination of Wedin's theorem [34] and Mirsky's inequality [23] implies

$$\|\boldsymbol{u}_i - \tilde{\boldsymbol{u}}_i\| \leq 2\sqrt{2}\|\boldsymbol{E}\|/\gamma_0. \tag{6}$$

This result provides an upper-bound on the change of every left singular vectors of the Jacobian.

However as we discuss below, data augmentation affects larger and smaller singular directions differently. To see the effect of data augmentation on every singular vectors of the Jacobian, let the subspace decomposition of Jacobian be $\mathcal{J} = \boldsymbol{U}\boldsymbol{\Sigma}\boldsymbol{V}^T = \boldsymbol{U}_s\boldsymbol{\Sigma}_s\boldsymbol{V}_s^T + \boldsymbol{U}_n\boldsymbol{\Sigma}_n\boldsymbol{V}_n^T$, where $\boldsymbol{U}_s$ associated with nonzero singular values, spans the column space of $\mathcal{J}$, which is also called the signal subspace, and $\boldsymbol{U}_n$, associated with zero singular values ($\boldsymbol{\Sigma}_n = 0$), spans the orthogonal space of $\boldsymbol{U}_s$, which is also called the noise subspace. Similarly, let the subspace decomposition of the perturbed Jacobian be $\tilde{\mathcal{J}} = \tilde{\boldsymbol{U}}\tilde{\boldsymbol{\Sigma}}\tilde{\boldsymbol{V}}^T = \tilde{\boldsymbol{U}}_s\tilde{\boldsymbol{\Sigma}}_s\tilde{\boldsymbol{V}}_s^T + \tilde{\boldsymbol{U}}_n\tilde{\boldsymbol{\Sigma}}_n\tilde{\boldsymbol{V}}_n^T$, and $\tilde{\boldsymbol{U}}_s = \boldsymbol{U}_s + \Delta\boldsymbol{U}_s$, where $\Delta\boldsymbol{U}_s$ is the perturbation of the singular vectors that span the signal subspace. Then the following general first-order expression for the perturbation of the orthogonal subspace due to perturbations of the Jacobian characterize the change of the singular directions: $\Delta\boldsymbol{U}_s = \boldsymbol{U}_n\boldsymbol{U}_n^T\boldsymbol{E}\boldsymbol{V}_s\boldsymbol{\Sigma}_s^{-1}$ [20]. We see that singular vectors associated to larger singular values are more robust to data augmentation, compared to others. Note that in general singular vectors are more robust than singular values.

Fig. 1 shows the effect of perturbations with $\epsilon_0 = 8, 16$ on singular values and singular vectors of the Jacobian matrix for a 1 hidden layer MLP trained on MNIST, and ResNet20 trained on CIFAR10. As calculating the entire Jacobian spectrum is computationally prohibitive, data is subsampled from 3 classes. We report the effect of other real-world augmentation techniques, such as random crops, flips, rotations and Autoaugment [6] - which includes translations, contrast, and brightness transforms - in Appendix C. We observe that data augmentation increases smaller singular values relatively more. On the other hand, it affects prominent singular vectors of the Jacobian to a smaller extent.

### 4.3 Augmentation Improves Training & Generalization

Recent studies have revealed that the Jacobian matrix of common neural networks is low rank. That is there are a number of large singular values and the rest of the singular values are small. Based on this, the Jacobian spectrum can be divided into information and nuisance spaces [27]. Information space is a lower dimensional space associated with the prominent singular value/vectors of the Jacobian. Nuisance space is a high dimensional space corresponding to smaller singular value/vectors of the Jacobian. While learning over information space is fast and generalizes well, learning over nuisance space is slow and results in overfitting [27]. Importantly, recent theoretical studies connected the generalization performance to small singular values (of the information space) [1].

Our results show that label-preserving additive perturbations relatively enlarge the smaller singular values of the Jacobian in a *stochastic* way and with a high probability. This benefits generalization in 2 ways. First, this stochastic behavior prevents overfitting along any particular singular direction *in the nuisance space*, as stochastic perturbation of the *smallest* singular values results in a stochastic noise to be added to the gradient at every training iteration. This prevents overfitting (thus a larger training loss as shown in Appendix D.5), and improves generalization [7, 8]. Theorem B.1 in the Appendix characterizes the expected training dynamics resulted by data augmentation. Second, additive perturbations improve the generalization by enlarging the smaller (useful) singular values that lie in the *information space*, while preserving eigenvectors. Hence, it enhances learning along these (harder to learn) components. The following Lemma captures the improvement in the generalization performance, as a result of data augmentation.

**Lemma 4.2.** *Assume gradient descent with learning rate $\eta$ is applied to train a neural network with constant NTK and Lipschitz constant L, on data points augmented with additive perturbations bounded by $\epsilon_0$ as defined in Sec. 3. Let $\sigma_{\min}$ be the minimum singular value of Jacobian $\mathcal{J}$ associated with training data $\boldsymbol{X}_{train}$. With probability $1 - \delta$, generalization error of the network trained with gradient descent on augmented data $\boldsymbol{X}_{aug}$ enjoys the following bound:*

$$\sqrt{\frac{2}{(\sigma_{\min} + \sqrt{n}L\epsilon_0)^2}} + \mathcal{O}\left(\log\frac{1}{\delta}\right). \tag{7}$$

The proof can be found in Appendix A.2.

## 5 Effective Subsets for Data Augmentation

Here, we focus on identifying subsets of data that when augmented similarly improve generalization and prevent overfitting. To do so, our key idea is to find subsets of data points that when augmented, closely capture the alignment of the NTK (or equivalently the Jacobian) corresponding to the full augmented data with the residual vector, $\mathcal{J}(\boldsymbol{W}^t, \boldsymbol{X}_{aug}^t)^T \boldsymbol{r}_{aug}^t$. If such subsets can be found, augmenting only the subsets will change the NTK and its alignment with the residual in a similar way as that of full data augmentation, and will result in similar improved training dynamics. However, generating the full set of transformations $\boldsymbol{X}_{aug}^t$ is often very expensive, particularly for strong augmentations and large datasets. Hence, generating the transformations, and then extracting the subsets may not provide a considerable overall speedup.

In the following, we show that weighted subsets (coresets) $S$ that closely estimate the alignment of the Jacobian associated to the original data with the residual vector $\mathcal{J}^T(\boldsymbol{W}^t, \boldsymbol{X}_{train})\boldsymbol{r}_{train}$ can closely estimate the alignment of the Jacobian of the full augmented data and the corresponding residual $\mathcal{J}^T(\boldsymbol{W}^t, \boldsymbol{X}_{aug}^t)\boldsymbol{r}_{aug}^t$. Thus, the most effective subsets for augmentation can be directly found from the training data. Formally, subsets $S_*^t$ weighted by $\boldsymbol{\gamma}_S^t$ that capture the alignment of the full Jacobian

---
**Algorithm 1** CORESETS FOR EFFICIENT DATA AUGMENTATION
---
**Require:** The dataset $\mathcal{D} = \{(\boldsymbol{x}_i, y_i)\}_{i=1}^n$, number of iterations $T$.
**Ensure:** Output model parameters $\boldsymbol{W}^T$.
1: **for** $t = 1, \cdots, T$ **do**
2:     $\boldsymbol{X}_{aug}^t = \emptyset$.
3:     **for** $c \in \{1, \cdots, C\}$ **do**
4:         $S_c^t = \emptyset$, $[\boldsymbol{G}_{S_c^t}]_{i.} = c_1 \boldsymbol{1}$  $\forall i$.
5:         **while** $\|\boldsymbol{G}_{S_c^t}\|_F \geq \xi$ **do**              ▷ Extract a coreset from class $c$ by solving Eq. (9)
6:             $S_c^t = \{S_c^t \cup \arg\max_{s \in V \setminus S_c^t} (\|\boldsymbol{G}_{S_c^t}\|_F - \|\boldsymbol{G}_{\{S_c^t \cup \{s\}\}}\|_F)\}$
7:         **end while**
8:         $\gamma_j = \sum_{i \in V_c} \mathbb{I}[j = \arg\min_{j' \in S} \|\mathcal{J}^T(\boldsymbol{W}^t, \boldsymbol{x}_i)r_i - \mathcal{J}^T(\boldsymbol{W}^t, \boldsymbol{x}_{j'})r_{j'}\|]$  ▷ Coreset weights
9:         $\boldsymbol{X}_{aug}^t = \{\boldsymbol{X}_{aug} \cup \{\cup_{i=1}^r T_i^t(\boldsymbol{X}_{S_c^t})\}\}$                ▷ Augment the coreset
10:        $\boldsymbol{\rho}_j^t = \gamma_j^t / r$
11:     **end for**
12:     Update the parameters $\boldsymbol{W}^t$ using weighted gradient descent on $\boldsymbol{X}_{aug}^t$ or $\{\boldsymbol{X}_{train} \cup \boldsymbol{X}_{aug}^t\}$.
13: **end for**
---

with the residual by an error of at most $\xi$ can be found by solving the following optimization problem:

$$S_*^t = \arg\min_{S \subseteq V} |S| \qquad \text{s.t.} \qquad \|\mathcal{J}^T(\boldsymbol{W}^t, \boldsymbol{X}^t)\boldsymbol{r}^t - \text{diag}(\boldsymbol{\gamma}_S^t)\mathcal{J}^T(\boldsymbol{W}^t, \boldsymbol{X}_S^t)\boldsymbol{r}_S^t\| \leq \xi. \qquad (8)$$

Solving the above optimization problem is NP-hard. However, as we discuss in the Appendix A.5, a near optimal subset can be found by minimizing the Frobenius norm of a matrix $\boldsymbol{G}_S$, in which the $i^{th}$ row contains the euclidean distance between data point $i$ and its closest element in the subset $S$, in the gradient space. Formally, $[\boldsymbol{G}_S]_{i.} = \min_{j' \in S} \|\mathcal{J}^T(\boldsymbol{W}^t, \boldsymbol{x}_i)r_i - \mathcal{J}^T(\boldsymbol{W}^t, \boldsymbol{x}_{j'})r_{j'}\|$. When $S = \emptyset$, $[\boldsymbol{G}_S]_{i.} = c_1 \boldsymbol{1}$, where $c_1$ is a big constant. Intuitively, such subsets contain the set of medoids of the dataset in the gradient space. Medoids of a dataset are defined as the most centrally located elements in the dataset [16]. The weight of every element $j \in S$ is the number of data points closest to it in the gradient space, i.e., $\gamma_j = \sum_{i \in V} \mathbb{I}[j = \arg\min_{j' \in S} \|\mathcal{J}^T(\boldsymbol{W}^t, \boldsymbol{x}_i)r_i - \mathcal{J}^T(\boldsymbol{W}^t, \boldsymbol{x}_{j'})r_{j'}\|]$. The set of medoids can be found by solving the following *submodular*[3] cover problem:

$$S_*^t = \arg\min_{S \subseteq V} |S| \quad s.t. \quad \|\boldsymbol{G}_S\|_F \leq \xi. \qquad (9)$$

The classical greedy algorithm provides a logarithmic approximation for the above submodular maximization problem, i.e., $|S| \leq (1 + ln(n))$. It starts with the empty set $S_0 = \emptyset$, and at each iteration $\tau$, it selects the training example $s \in V \setminus S_{\tau-1}$ that maximizes the marginal gain, i.e., $S_\tau = S_{\tau-1} \cup \{\arg\max_{s \in V \setminus S_{\tau-1}} (\|\boldsymbol{G}_{S_{\tau-1}}\|_F - \|\boldsymbol{G}_{\{S_{\tau-1} \cup \{s\}\}}\|_F)\}$. The $\mathcal{O}(nk)$ computational complexity of the greedy algorithm can be reduced to $\mathcal{O}(n)$ using randomized methods [25] and further improved using lazy evaluation [22] and distributed implementations [26]. The rows of the matrix $\boldsymbol{G}$ can be efficiently upper-bounded using the gradient of the loss w.r.t. the input to the last layer of the network, which has been shown to capture the variation of the gradient norms closely [15]. The above upper-bound is only marginally more expensive than calculating the value of the loss. Hence the subset can be found efficiently. Better approximations can be obtained by considering earlier layers in addition to the last two, at the expense of greater computational cost.

At every iteration $t$ during training, we select a coreset from every class $c \in [C]$ separately, and apply the set of transformations $\{T_i^t\}_{i=1}^r$ only to the elements of the coresets, i.e., $X_{aug}^t = \{\cup_{i=1}^r T_i^t(\boldsymbol{X}_{S^t})\}$. We divide the weight of every element $j$ in the coreset equally among its transformations, i.e. the final weight $\rho_j^t = \gamma_j^t / r$ if $j \in S^t$. We apply the gradient descent updates in Eq. (3) to the weighted Jacobian matrix of $\boldsymbol{X}^t = \boldsymbol{X}_{aug}^t$ or $\boldsymbol{X}^t = \{\boldsymbol{X}_{train} \cup \boldsymbol{X}_{aug}^t\}$ (viewing $\boldsymbol{\rho}^t$ as $\boldsymbol{\rho}^t \in \mathbb{R}^n$) as follows:

$$\boldsymbol{W}^{t+1} = \boldsymbol{W}^t - \eta \left(\text{diag}(\boldsymbol{\rho}^t)\mathcal{J}(\boldsymbol{W}^t, \boldsymbol{X}^t)\right)^T \boldsymbol{r}^t. \qquad (10)$$

The pseudocode is illustrated in Alg. 1.

The following Lemma upper bounds the difference between the alignment of the Jacobian and residual for augmented coreset vs. full augmented data.

---
[3] A set function $F : 2^V \to \mathbb{R}^+$ is submodular if $F(S \cup \{e\}) - F(S) \geq F(T \cup \{e\}) - F(T)$, for any $S \subseteq T \subseteq V$ and $e \in V \setminus T$. $F$ is *monotone* if $F(e|S) \geq 0$ for any $e \in V \setminus S$ and $S \subseteq V$.

Table 1: Training ResNet20 (R20) and WideResnet-28-10 (W2810) on CIFAR10 (C10) using small subsets, and ResNet18 (R18) on Caltech256 (Cal). We compare accuracies of training on and strongly (and weakly) augmenting subsets. For CIFAR10, training and augmenting subsets selected by max-loss performed poorly and did not converge. Average number of examples per class in each subset is shown in parentheses. Appendix D.4 shows baseline accuracies from only weak augmentations.

| Model/Data | C10/R20 | | | | C10/W2810 | Cal/R18 | | | | | |
|---|---|---|---|---|---|---|---|---|---|---|---|
| Subset | 0.1% (5) | 0.2% (10) | 0.5% (25) | 1% (50) | 1% (50) | 5% (3) | 10% (6) | 20% (12) | 30% (18) | 40% (24) | 50% (30) |
| Max-loss | $< 15\%$ | $< 15\%$ | $< 15\%$ | $< 15\%$ | $< 15\%$ | 19.2 | 50.6 | 71.3 | 75.6 | 77.3 | 78.6 |
| Random | 33.5 | 42.7 | 58.7 | 74.4 | 57.7 | 41.5 | 61.8 | 72.5 | 75.7 | 77.6 | 78.5 |
| Ours | **37.8** | **45.1** | **63.9** | **74.7** | **62.1** | **52.7** | **65.4** | **73.1** | **76.3** | **77.7** | **78.9** |

**Lemma 5.1.** *Let $S$ be a coreset that captures the alignment of the full data NTK with residual with an error of at most $\xi$ as in Eq. 8. Augmenting the coreset with perturbations bounded by $\epsilon_0 \leq \frac{1}{n^{\frac{3}{2}}\sqrt{L}}$ captures the alignment of the fully augmented data with the residual by an error of at most*

$$\|\mathcal{J}^T(\boldsymbol{W}^t, \boldsymbol{X}_{aug})\boldsymbol{r} - diag(\boldsymbol{\rho}^t)\mathcal{J}^t(\boldsymbol{W}^t, \boldsymbol{X}_{S^{aug}})\boldsymbol{r}_S\| \leq \xi + \mathcal{O}\left(\sqrt{L}\right). \quad (11)$$

### 5.1 Coreset vs. Max-loss Data Augmentation

In the initial phase of training the NTK goes through rapid changes. This determines the final basin of convergence and network's final performance [10]. Regularizing deep networks by weight decay or data augmentation mainly affects this initial phase and matters little afterwards [11]. Crucially, augmenting coresets that closely capture the alignment of the NTK with the residual during this initial phase results in less overfitting and improved generalization performance. On the other hand, augmenting points with maximum loss early in training decreases the alignment between the NTK and the label vector and impedes learning and convergence. After this initial phase when the network has good prediction performance, the gradients for majority of data points become small. Here, the alignment is mainly captured by the elements with the maximum loss. Thus, as training proceeds, the intersection between the elements of the coresets and examples with maximum loss increases. We visualize this pattern in Appendix D.11. The following Theorem characterizes the training dynamics of training on the full data and the augmented coresets, using our additive perturbation model.

**Theorem 5.2.** *Let $\mathcal{L}_i$ be $\beta$-smooth, $\mathcal{L}$ be $\lambda$-smooth and satisfy the $\alpha$-PL condition, that is for $\alpha > 0$, $\|\nabla\mathcal{L}(\boldsymbol{W})\|^2 \geq \alpha\mathcal{L}(\boldsymbol{W})$ for all weights $\boldsymbol{W}$. Let $f$ be Lipschitz in $\boldsymbol{X}$ with constant $L'$, and $\bar{L} = \max\{L, L'\}$. Let $G_0$ be the gradient at initializaion, $\sigma_{\max}$ the maximum singular value of the coreset Jacobian at initialization. Choosing $\epsilon_0 \leq \frac{1}{\sigma_{\max}\sqrt{\bar{L}n}}$ and running SGD on full data with augmented coreset using constant step size $\eta = \frac{\alpha}{\lambda\beta}$, result in the following bound:*

$$\mathbb{E}[\|\nabla\mathcal{L}^{f+c_{\text{aug}}}(\boldsymbol{W}^t)\|] \leq \frac{1}{\sqrt{\alpha}}\left(1 - \frac{\alpha\eta}{2}\right)^{\frac{t}{2}}\left(2G_0 + \xi + \mathcal{O}\left(\frac{\sqrt{\bar{L}}}{\sigma_{\max}}\right)\right).$$

The proof can be found in Appendix A.4.

Theorem 5.2 shows that training on full data and augmented coresets converges to a close neighborhood of the optimal solution, with the same rate as that of training on the fully augmented data. The size of the neighborhood depends on the error of the coreset $\xi$ in Eq. (8), and the error in capturing the alignment of the full augmented data with the residual derived in Lemma 5.1. The first term decrease as the size of the coreset grows, and the second term depends on the network structure.

We also analyze convergence of training only on the augmented coresets, and augmentations modelled as arbitrary linear transformations using a linear model [36] in Appendix B.5.

## 6 Experiments

**Setup and baselines.** We extensively evaluate the performance of our approach in three different settings. Firstly, we consider training only on coresets and their augmentations. Secondly, we investigate the effect of adding augmented coresets to the full training data. Finally, we consider adding augmented coresets to random subsets. We compare our coresets with max-loss and random subsets as baselines. For all methods, we select a new augmentation subset every $R$ epochs. We note that the original max-loss method [17] selects points using a fully trained model, hence it can only

Table 2: Caltech256/ResNet18 with same settings as Tab. 1 with default weak augmentations but varying strong augmentations.

| Augmentation | Random | | | Ours | | |
|---|---|---|---|---|---|---|
| | 30% | 40% | 50% | 30% | 40% | 50% |
| CutOut | 43.32 | 62.84 | 76.21 | **55.53** | **66.10** | **76.91** |
| AugMix | 40.77 | 61.81 | 72.17 | **52.72** | **64.91** | **73.01** |
| Perturb | 48.51 | 66.20 | 75.34 | **58.29** | **67.47** | **76.50** |

Table 3: Training on full data and strongly (and weakly) augmenting random subsets, max-loss subsets and coresets on TinyImageNet/ResNet50, $R = 15$.

| Random | | | Max-loss | | | Ours | | |
|---|---|---|---|---|---|---|---|---|
| 20% | 30% | 50% | 20% | 30% | 50% | 20% | 30% | 50% |
| 50.97 | 52.00 | 54.92 | 51.30 | 52.34 | 53.37 | **51.99** | **54.30** | **55.16** |

Table 4: Accuracy improvement by augmenting subsets found by our method vs. max-loss and random, over improvement of full (weak and strong) data augmentation (F.A.) compared to weak augmentation only (W.A.). The table shows the results for training on CIFAR10(C10)/ResNet20 (R20), SVHN/ResNet32(R32), and CIFAR10-Imbalanced(C10-IB)/ResNet32, with $R = 20$.

| Dataset | W.A. | F.A. | Random | | | Max-loss | | | Ours | | |
|---|---|---|---|---|---|---|---|---|---|---|---|
| | Acc | Acc | 5% | 10% | 30% | 5% | 10% | 30% | 5% | 10% | 30% |
| C10/R20 | 89.46 | 93.50 | 21.8% | 39.9% | 65.6% | 32.9% | 47.8% | 73.5% | **34.9%** | **51.5%** | **75.0%** |
| C10-IB/R32 | 87.08 | 92.48 | 25.9% | 45.2% | 74.6% | 31.3% | 39.6% | 74.6% | **37.4%** | **49.4%** | **74.8%** |
| SVHN/R32 | 95.68 | 97.07 | 5.8% | 36.7% | 64.1% | **35.3%** | **49.7%** | 76.4% | 31.7% | 48.3% | **80.0%** |

select one subset throughout training. To maximize fairness, we modify our max-loss baseline to select a new subset at every subset selection step. For all experiments, standard weak augmentations (random crop and horizontal flips) are always performed on both the original and strongly augmented data.

## 6.1 Training on Coresets and their Augmentations

First, we evaluate the effectiveness of our approach for training on the coresets and their augmentations. Our main goal here is to compare the performance of training on and augmenting coresets vs. random and max-loss subsets. Tab. 1 shows the test accuracy for training ResNet20 and Wide-ResNet on CIFAR10 when we only train on small augmented coresets of size $0.1\%$ to $1\%$ selected at every epoch ($R = 1$), and training ResNet18 on Caltech256 using coresets of size $5\%$ to $50\%$ with $R = 5$. We see that the augmented coresets outperform augmented random subsets by a large margin, particularly when the size of the subset is small. On Caltech256/ResNet18, training on and augmenting 10% coresets yields 65.4% accuracy, improving over random by 5.8%, and over only weak augmentation by 17.4%. This clearly shows the effectiveness of augmenting the coresets. Note that for CIFAR10 experiments, training on the augmented max-loss points did not even converge in absence of full data.

**Generalization across augmentation techniques.** We note that our coresets are not dependent on the type of data augmentation. To confirm this, we show the superior generalization performance of our method in Tab. 2 for training ResNet18 with $R = 5$ on coresets vs. random subsets of Caltech256, augmented with CutOut [9], AugMix [12], and noise perturbations (color jitter, gaussian blur). For example, on 30% subsets, we obtain 28.2%, 29.3%, 20.2% relative improvement over augmenting random subsets when using CutOut, AugMix, and noise perturbation augmentations, respectively.

## 6.2 Training on Full Data and Augmented Coresets

Next, we study the effectiveness of our method for training on full data and augmented coresets. Tab. 4 demonstrates the percentage of accuracy improvement resulted by augmenting subsets of size 5%, 10%, and 30% selected from our method vs. max-loss and random subsets, over that of full data augmentation. We observe that augmenting coresets effectively improves generalization, and outperforms augmenting random and max-loss subsets across different models and datasets. For example, on 30% subsets, we obtain 13.1% and 2.3% relative improvement over random and max-loss on average. We also report results on TinyImageNet/ResNet50 ($R = 15$) in Tab. 3, where we show that augmenting coresets outperforms max-loss and random baselines, e.g. by achieving 3.7% and 4.4% relative improvement over 30% max-loss and random subsets, respectively.

**Training speedup.** In Fig. 2, we measure the improvement in training time in the case of training on full data and augmenting subsets of various sizes. While our method yields similar or slightly lower speedup to the max-loss and random baselines, our resulting accuracy outperforms the baselines on average. For example, for SVHN/Resnet32 using 30% coresets, we sacrifice 11% of the relative speedup to obtain an additional 24.8% of the relative gain in accuracy from full data augmentation, compared to random baseline. Notably, we get 3.43x speedup for training on full data and augmenting

Table 5: Training ResNet20 on CIFAR10 with $50\%$ label noise, $R = 20$. Accuracy without strong augmentation is $70.72 \pm 0.20$ and the accuracy of full (weak and strong) data augmentation is $75.87 \pm 0.77$. Note that augmenting $50\%$ subsets outperforms augmenting the full data (marked **).

| Subset | Random | Max-loss | Ours |
|--------|--------|----------|------|
| 10% | $72.32 \pm 0.14$ | $71.83 \pm 0.13$ | $\mathbf{73.02 \pm 1.06}$ |
| 30% | $74.46 \pm 0.27$ | $72.45 \pm 0.48$ | $\mathbf{74.67 \pm 0.15}$ |
| 50% | $75.36 \pm 0.05$ | $73.23 \pm 0.72$ | $\mathbf{76.20 \pm 0.75}$** |

30% coresets, while obtaining 75% of the improvement of full data augmentation. We provide wall-clock times for finding coresets from Caltech256 and TinyImageNet in Appendix D.7.

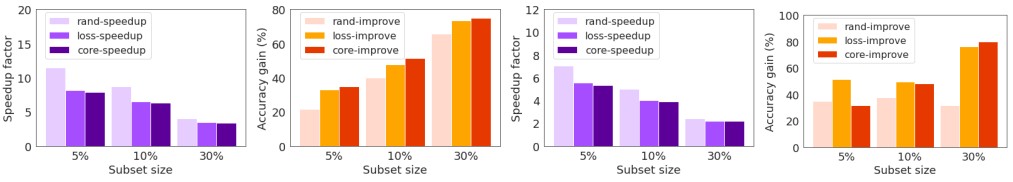

|(a) CIFAR10/ResNet20 | (b) CIFAR10/ResNet20 | (c) SVHN/ResNet32 | (d) SVHN/ResNet32|

Figure 2: Accuracy improvement and speedups by augmenting subsets found by our method vs. max-loss and random on (a), (b) ResNet20/CIFAR10 and (c), (d) ResNet32/SVHN.

**Augmenting noisy labeled data.** Next, we evaluate the robustness of our coresets to label noise. Tab. 5 shows the result of augmenting coresets vs. max-loss and random subsets of different sizes selected from CIFAR10 with 50% label noise on ResNet20. Notably, our method not only outperforms max-loss and random baselines, but also achieves superior performance over full data augmentation.

### 6.3 Training on Random Data and Augmented Coresets

Finally, we evaluate the performance of our method for training on random subsets and augmenting coresets, applicable when data is larger than the training budget. We report results on TinyImageNet and ImageNet on ResNet50 (90 epochs, $R = 15$). Tab. 6 shows the results of training on random subsets, and augmenting random subsets and coresets of the same size. We see that our results hold for large-scale datasets, where we obtain 7.9%, 4.9%, and 5.3% relative improvement over random baseline with 10%, 20%, 30% subset sizes respectively on TinyImageNet, and 7.6%, 2.3%, and 1.3% relative improvement over random baseline with 10%, 30%, and 50% subset sizes on ImageNet. Notably, compared to AutoAugment, despite using only 30% subsets, we achieve 71.99% test accuracy, which is 92.8% of the original reported accuracy, while boasting 5x speedup in training.

Table 6: Training on random subsets and strongly (and weakly) augmenting random and max loss subsets vs coresets for TinyImageNet (left) and ImageNet (right) with ResNet50.

| Random | | | Max-loss | | | Ours | | | Random | | | Maxloss | | | Ours | | |
|--------|--------|--------|--------|--------|--------|------|------|------|--------|--------|--------|--------|--------|--------|------|------|------|
| 10% | 20% | 30% | 10% | 20% | 30% | 10% | 20% | 30% | 10% | 30% | 50% | 10% | 30% | 50% | 10% | 30% | 50% |
| 28.64 | 38.97 | 44.10 | 27.64 | **41.40** | 45.75 | **30.90** | 40.88 | **46.42** | 63.67 | 70.39 | 72.35 | 65.43 | 71.55 | 72.77 | **68.53** | **71.99** | **73.28** |

## 7 Conclusion

We showed that data augmentation improves training and generalization by relatively enlarging and perturbing the smaller singular values of the neural network Jacobian while preserving its prominent directions. Then, we proposed a framework to iteratively extract small coresets of training data that when augmented, closely capture the alignment of the fully augmented Jacobian with the label/residual vector. We showed the effectiveness of augmenting coresets in providing a superior generalization performance when added to the full data or random subsets, in presence of noisy labels, or as a standalone subset. Under local smoothness of images, our additive perturbation can be applied to model many bounded transformations such as small rotations, crops, shearing, and pixel-wise transformations like sharpening, blurring, color distortions, structured adversarial perturbation [21]. However, the additive perturbation model is indeed limited when applied to augmentations that cannot be reduced to perturbations, such as horizontal/vertical flips and large translations. Further theoretical analysis of complex data augmentations is indeed an interesting direction for future work.

## 8 Acknowledgements

This research was supported in part by the National Science Foundation CAREER Award 2146492, and the UCLA-Amazon Science Hub for Humanity and AI.

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
