# Supplementary Material: Data-Efficient Augmentation for Training Neural Networks

## A Proof of Main Results

### A.1 Proof for Lemma 4.1

*Proof.* Let $\delta_i := \tilde{\sigma}_i - \sigma_i$, where $\mathbb{P}(\delta_i < 0) = p_i$. Assuming uniform probability between $-\|\boldsymbol{E}\|$ to 0, and between 0 to $\|\boldsymbol{E}\|$, we have pdf $\rho_i(x)$ for $\delta_i$:

$$\rho_i(x) = \begin{cases} \frac{p_i}{\|\boldsymbol{E}\|}, & \text{if } -\|\boldsymbol{E}\| \leq x < 0 \\ \frac{1-p_i}{\|\boldsymbol{E}\|}, & 0 \leq x \leq \|\boldsymbol{E}\| \\ 0, & \text{otherwise} \end{cases} \tag{1}$$

Taking expectation,

$$\mathbb{E}(\tilde{\sigma}_i - \sigma_i) = \mathbb{E}(\delta_i) = \int_{-\infty}^{\infty} x\rho_i(x)dx \tag{2}$$

$$= \int_{-\|\boldsymbol{E}\|}^{0} x\frac{p_i}{\|\boldsymbol{E}\|}dx + \int_{0}^{\|\boldsymbol{E}\|} x\frac{1-p_i}{\|\boldsymbol{E}\|}dx \tag{3}$$

$$= -\frac{\|\boldsymbol{E}\|p_i}{2} + \frac{(1-p_i)\|\boldsymbol{E}\|}{2} \tag{4}$$

$$= \frac{(1-2p_i)\|E\|}{2} \tag{5}$$

We also have

$$\mathbb{E}(\delta_i^2) = \int_{-\infty}^{\infty} x^2\rho_i(x)dx \tag{6}$$

$$= \int_{-\|\boldsymbol{E}\|}^{0} x^2\frac{p_i}{\|\boldsymbol{E}\|}dx + \int_{0}^{\|\boldsymbol{E}\|} x^2\frac{1-p_i}{\|\boldsymbol{E}\|}dx \tag{7}$$

$$= \frac{\|\boldsymbol{E}\|^2 p_i}{3} + \frac{(1-p_i)\|\boldsymbol{E}\|^2}{3} \tag{8}$$

$$= \frac{\|\boldsymbol{E}\|^2}{3} \tag{9}$$

Thus, we have

$$\mathbb{E}(\tilde{\lambda}_i) = \mathbb{E}(\tilde{\sigma}_i^2) \tag{10}$$

$$= \mathbb{E}((\sigma_i + \delta_i)^2) \tag{11}$$

$$= \mathbb{E}(\sigma_i^2 + 2\sigma_i\delta_i + \delta_i^2) \tag{12}$$

$$= \sigma_i^2 + 2\sigma_i\mathbb{E}[\delta_i] + \mathbb{E}[\delta_i^2] \tag{13}$$

$$= \sigma_i^2 + 2\sigma_i\frac{(1-2p_i)\|\boldsymbol{E}\|}{2} + \frac{\|\boldsymbol{E}\|^2}{3} \tag{14}$$

$$= \sigma_i^2 + \sigma_i(1-2p_i)\|\boldsymbol{E}\| + \frac{\|\boldsymbol{E}\|^2}{3} \tag{15}$$

$\square$

## A.2 Proof of Corollary 4.2

Under the assumptions of Theorem 5.1 of [1], i.e. where the minimum eigenvalue of the NTK is $\lambda_{\min}(\mathcal{J}\mathcal{J}^T) \geq \lambda_0$ for a constant $\lambda_0 > 0$, and training data $\boldsymbol{X}$ of size $n$ sampled i.i.d. from distribution $D$ and 1-Lipschitz loss $\mathcal{L}$, we have that with probability $\delta/3$, training the over-parameterized neural network with gradient descent for $t \geq \Omega\left(\frac{1}{n\lambda_0}\log\frac{n}{\delta}\right)$ iterations results in the following population loss $\mathcal{L}_D$ (generalization error)

$$\mathcal{L}_D(\boldsymbol{W}^t, \boldsymbol{X}) \leq \sqrt{\frac{2\boldsymbol{y}^T(\mathcal{J}\mathcal{J}^T)^{-1}\boldsymbol{y}}{n}} + \mathcal{O}\left(\frac{\log\frac{n}{\lambda_0\delta}}{n}\right), \tag{16}$$

with high probability of at least $1 - \delta$ over random initialization and training samples.

Hence, using $\lambda_{\min}, \sigma_{\min}$ to denote minimum eigen and singular value respectively of the NTK corresponding to full data, we get

$$\mathcal{L}_{D_{train}}(\boldsymbol{W}^t, \boldsymbol{X}_{train}) \leq \sqrt{\frac{2\frac{1}{\lambda_{\min}}\|y\|^2}{n}} + \mathcal{O}\left(\log\frac{1}{\delta}\right) \tag{17}$$

$$\leq \sqrt{\frac{2}{\sigma_{\min}^2}} + \mathcal{O}\left(\log\frac{1}{\delta}\right). \tag{18}$$

For augmented dataset $\boldsymbol{X}_{aug}$, we have $\tilde{\sigma}_i \leq \sigma_i + \sqrt{n}L\epsilon_0$, hence the improvement in the generalization error is at most

$$\mathcal{L}_{D_{aug}}(\boldsymbol{W}^t, \boldsymbol{X}_{aug}) \leq \sqrt{\frac{2}{(\sigma_{\min} + \sqrt{n}L\epsilon_0)^2}} + \mathcal{O}\left(\log\frac{1}{\delta}\right). \tag{19}$$

Combining these two results, we obtain Corollary 4.2.

## A.3 Proof of Lemma 5.1

*Proof.*
$$\|\mathcal{J}^T(\boldsymbol{W}^t, \boldsymbol{X}_{aug})\boldsymbol{r} - \text{diag}(\boldsymbol{\rho}^t)\mathcal{J}^t(\boldsymbol{W}^t, \boldsymbol{X}_{S^{aug}})\boldsymbol{r}_S\| \tag{20}$$

$$= \|(\mathcal{J}^T(\boldsymbol{W}^t, \boldsymbol{X}) + \boldsymbol{E})\boldsymbol{r} - (\text{diag}(\boldsymbol{\rho}^t)\mathcal{J}^t(\boldsymbol{W}^t, \boldsymbol{X}_S) + \boldsymbol{E}_S)\boldsymbol{r}_S\| \tag{21}$$

$$\leq \|(\mathcal{J}^T(\boldsymbol{W}^t, \boldsymbol{X})\boldsymbol{r} - \text{diag}(\boldsymbol{\rho}^t)\mathcal{J}^t(\boldsymbol{W}^t, \boldsymbol{X}_S)\boldsymbol{r}_S) + \boldsymbol{E}\boldsymbol{r} - \boldsymbol{E}_S\boldsymbol{r}_S\| \tag{22}$$

$$\leq \|(\mathcal{J}^T(\boldsymbol{W}^t, \boldsymbol{X})\boldsymbol{r} - \text{diag}(\boldsymbol{\rho}^t)\mathcal{J}^t(\boldsymbol{W}^t, \boldsymbol{X}_S)\boldsymbol{r}_S)\| + \|\boldsymbol{E}\boldsymbol{r}\| + \|\boldsymbol{E}_S\boldsymbol{r}_S\| \tag{23}$$

Applying definition of coresets, we obtain
$$\|(\mathcal{J}^T(\boldsymbol{W}^t, \boldsymbol{X})\boldsymbol{r} - \text{diag}(\boldsymbol{\rho}^t)\mathcal{J}^t(\boldsymbol{W}^t, \boldsymbol{X}_S)\boldsymbol{r}_S)\| + \|\boldsymbol{E}\boldsymbol{r}\| + \|\boldsymbol{E}_S\boldsymbol{r}_S\| \tag{24}$$

$$\leq \xi + \|\boldsymbol{E}\boldsymbol{r}\| + \|\boldsymbol{E}_S\boldsymbol{r}_S\| \tag{25}$$

$$\leq \xi + 2n^{\frac{3}{2}}L\epsilon_0 \tag{26}$$

$$\leq \xi + 2\sqrt{L} \tag{27}$$

$\square$

## A.4 Proof of Theorem 5.2

*Proof.* In this proof, as shorthand notation, we use $\boldsymbol{X}_f$ and $\boldsymbol{X}_{train}$ interchangeably. We further use $\boldsymbol{X}_c$ to represent the coreset selected from the full data, and $\boldsymbol{X}_{c_{aug}}$ to represent the augmented coreset.

By Theorem 1 of [2], under the $\alpha$-PL assumption for $\mathcal{L}$ and interpolation assumption (i.e. for every sequence $\boldsymbol{W}^1, \boldsymbol{W}^2, \ldots$ such that $\lim_{t\to\infty}\mathcal{L}(\boldsymbol{W}^t, \boldsymbol{X}) = 0$, we have that the loss for each data point $\lim_{t\to\infty}\mathcal{L}(\boldsymbol{W}^t, \boldsymbol{x}_i) = 0$), the convergence of SGD with constant step size is given by

$$\mathbb{E}[\|\nabla\mathcal{L}(\boldsymbol{W}^t, \boldsymbol{X}_{f+c_{aug}})\|^2] \leq \left(1 - \frac{\alpha\eta}{2}\right)^t \mathcal{L}(\boldsymbol{W}^0, \boldsymbol{X}_{f+c_{aug}}) \tag{28}$$

$$\leq \frac{1}{\alpha}\left(1 - \frac{\alpha\eta}{2}\right)^t \|\nabla\mathcal{L}(\boldsymbol{W}^0, \boldsymbol{X}_{f+c_{aug}})\|^2 \tag{29}$$

Using Jensen's inequality, we have

$$\mathbb{E}[\|\nabla\mathcal{L}(\boldsymbol{W}^0, \boldsymbol{X}_{f+c_{\mathrm{aug}}})\||] \tag{30}$$

$$\leq \sqrt{\mathbb{E}[\|\nabla\mathcal{L}(\boldsymbol{W}^t, \boldsymbol{X}_{f+c_{\mathrm{aug}}})\|^2]} \tag{31}$$

$$\leq \frac{1}{\sqrt{\alpha}}\left(1 - \frac{\alpha\eta}{2}\right)^{\frac{t}{2}} \|\nabla\mathcal{L}(\boldsymbol{W}^0, \boldsymbol{X}_{f+c_{\mathrm{aug}}})\| \tag{32}$$

$$\leq \frac{1}{\sqrt{\alpha}}\left(1 - \frac{\alpha\eta}{2}\right)^{\frac{t}{2}} \left(\|\nabla\mathcal{L}(\boldsymbol{W}^0, \boldsymbol{X}_f)\| + \|\nabla\mathcal{L}(\boldsymbol{W}^0, \boldsymbol{X}_{c_{\mathrm{aug}}})\|\right) \tag{33}$$

$$\leq \frac{1}{\sqrt{\alpha}}\left(1 - \frac{\alpha\eta}{2}\right)^{\frac{t}{2}} \left(G_0 + \|(\mathcal{J}(\boldsymbol{W}^0, \boldsymbol{X}_c) + \boldsymbol{E})(\boldsymbol{y} - f(\boldsymbol{W}^0, \boldsymbol{X}_c + \boldsymbol{\epsilon}))\|\right) \tag{34}$$

$$\leq \frac{1}{\sqrt{\alpha}}\left(1 - \frac{\alpha\eta}{2}\right)^{\frac{t}{2}} \tag{35}$$

$$\left(G_0 + \|(\mathcal{J}(\boldsymbol{W}^0, \boldsymbol{X}_c) + \boldsymbol{E})^T(\boldsymbol{y} - f(\boldsymbol{W}^0, \boldsymbol{X}_c) - \nabla_x f(\boldsymbol{W}^0, \boldsymbol{X}_c)^T\boldsymbol{\epsilon} - \mathcal{O}(\boldsymbol{\epsilon}^T\boldsymbol{\epsilon})\|\right) \tag{36}$$

$$= \frac{1}{\sqrt{\alpha}}\left(1 - \frac{\alpha\eta}{2}\right)^{\frac{t}{2}} \left(G_0 + \|\nabla L(\boldsymbol{W}^0, \boldsymbol{X}_c) - (\mathcal{J}(\boldsymbol{W}^0, \boldsymbol{X}_c)^T(\nabla_x f(\boldsymbol{W}^0, \boldsymbol{X}_c)^T\boldsymbol{\epsilon} + \mathcal{O}(\boldsymbol{\epsilon}^T\boldsymbol{\epsilon})) + \right. \tag{37}$$

$$\left. \boldsymbol{E}(\boldsymbol{y} - f(\boldsymbol{W}^0, \boldsymbol{X}_c + \boldsymbol{\epsilon}))\|\right) \tag{38}$$

$$\leq \frac{1}{\sqrt{\alpha}}\left(1 - \frac{\alpha\eta}{2}\right)^{\frac{t}{2}} \left(G_0 + \|\nabla L(\boldsymbol{W}^0, \boldsymbol{X}_c) - (\mathcal{J}(\boldsymbol{W}^0, \boldsymbol{X}_c)^T(\nabla_x f(\boldsymbol{W}^0, \boldsymbol{X}_c)^T\boldsymbol{\epsilon} + \mathcal{O}(\boldsymbol{\epsilon}^T\boldsymbol{\epsilon}))\| + \right. \tag{39}$$

$$\left. \sqrt{2}\|\boldsymbol{E}\|\right) \tag{40}$$

$$\leq \frac{1}{\sqrt{\alpha}}\left(1 - \frac{\alpha\eta}{2}\right)^{\frac{t}{2}} \left(G_0 + \|\nabla L(\boldsymbol{W}^0, \boldsymbol{X}_c)\| + \sigma_{\max}\bar{L}\sqrt{n}\epsilon_0 + \sigma_{\max}\mathcal{O}(n\epsilon_0^2)) + \sqrt{2n}\bar{L}\epsilon_0\right) \tag{41}$$

$$= \frac{1}{\sqrt{\alpha}}\left(1 - \frac{\alpha\eta}{2}\right)^{\frac{t}{2}} \left(G_0 + \|\nabla L(\boldsymbol{W}^0, \boldsymbol{X}_f)\| + \xi + \sigma_{\max}\bar{L}\sqrt{n}\epsilon_0 + \sigma_{\max}\mathcal{O}(n\epsilon_0^2)) + \sqrt{2n}\bar{L}\epsilon_0\right) \tag{42}$$

$$\leq \frac{1}{\sqrt{\alpha}}\left(1 - \frac{\alpha\eta}{2}\right)^{\frac{t}{2}} \left(2G_0 + \xi + \sigma_{\max}\bar{L}\sqrt{n}\epsilon_0 + \sigma_{\max}\mathcal{O}(n\epsilon_0^2)) + \sqrt{2n}\bar{L}\epsilon_0\right) \tag{43}$$

$$\qquad\qquad\qquad\qquad\qquad\qquad\qquad\qquad\qquad\qquad\qquad\qquad\qquad\qquad \square$$

## A.5  Finding Subsets

Let $S$ be a subset of training data points. Furthermore, assume that there is a mapping $\pi_{w,S} : V \to S$ that for every $\boldsymbol{W}$ assigns every data point $i \in V$ to its closest element $j \in S$, i.e. $j = \pi_{w,S}(i) = \arg\max_{j' \in S} s_{ij'}(\boldsymbol{W})$, where $s_{ij}(\boldsymbol{W}) = C - \|\mathcal{J}^T(\boldsymbol{W}^t, \boldsymbol{x}_i)r_i - \mathcal{J}^T(\boldsymbol{W}^t, \boldsymbol{x}_j)r_j\|$ is the similarity between gradients of $i$ and $j$, and $C \geq \max_{ij} s_{ij}(\boldsymbol{W})$ is a constant. Consider a matrix $\boldsymbol{G}_{\pi_{w,S}} \in \mathbb{R}^{n \times m}$, in which every row $i$ contains gradient of $\pi_w(i)$, i.e., $[\boldsymbol{G}_{\pi_{w,S}}]_{i\cdot} = \mathcal{J}^T(\boldsymbol{W}^t, \boldsymbol{x}_{\pi_{w,S}(i)})r_{\pi_{w,S}(i)}$. The Frobenius norm of the matrix $\boldsymbol{G}_{\pi_w}$ provides an upper-bound on the error of the weighted subset $S$ in capturing the alignment of the residuals of the full training data with the Jacobian matrix. Formally,

$$\|\mathcal{J}^T(\boldsymbol{W}^t, \boldsymbol{X}_{train})\boldsymbol{r}_{train}^t - \boldsymbol{\gamma}_{S^t}\mathcal{J}^T(\boldsymbol{W}^t, [\boldsymbol{X}_{train}]_{.S^t})r_{S^t}\| \leq \|\boldsymbol{G}_{\pi_{w,S}}\|_F, \tag{44}$$

where the weight vector $\boldsymbol{\gamma}_{S^t} \in \mathbb{R}^{|S|}$ contains the number of elements that are mapped to every element $j \in S$ by mapping $\pi_{w,S}$, i.e. $\gamma_j = \sum_{i \in V} \mathbb{1}[\pi_{w,S}(i) = j]$. Hence, the set of training points that closely estimate the projection of the residuals of the full training data on the Jacobian spectrum can be obtained by finding a subset $S$ that minimizes the Frobenius norm of matrix $\boldsymbol{G}_{\pi_{w,S}}$.

## B  Additional Theoretical Results

### B.1  Convergence analysis for training on augmented full data

**Theorem B.1.** *Gradient descent with learning rate $\eta$ applied to a neural network with constant NTK and Lipschitz constant $L$, and data points $\mathcal{D}_{aug}$ augmented with $r$ additive perturbations bounded by $\epsilon_0$ results in the following training dynamics:*

$$\mathbb{E}[\|\boldsymbol{y} - f(\boldsymbol{X}_{aug}, \boldsymbol{W}^t)\|_2] \leq$$
$$\sqrt{\sum_{i=1}^{n} \left(1 - \eta\left(\sigma_i^2 + \sigma_i(1-2p_i)\|E\| + \frac{\|E\|^2}{3}\right)\right)^{2t} ((\boldsymbol{u}_i\boldsymbol{y})^2 + 2n\sqrt{2}\|E\|/\gamma_0)} \tag{45}$$

*where $\boldsymbol{E}$ with $\|\boldsymbol{E}\| \leq \sqrt{n}L\epsilon_0$ is the perturbation to the Jacobian, and $p_i := \mathbb{P}(\tilde{\sigma}_i - \sigma_i < 0)$ is the probability that $\sigma_i$ decreases as a result of data augmentation.*

### B.2  Proof of Theorem B.1

Using Jensen's inequality, we have

$$\mathbb{E}\left[\|\boldsymbol{y} - f(\boldsymbol{X}_{aug}, \boldsymbol{W}^t)\|_2\right] \tag{46}$$

$$= \mathbb{E}\left[\sqrt{\sum_{i=1}^{n}(1-\eta\tilde{\lambda}_i)^{2t}(\tilde{\boldsymbol{u}}_i^T\boldsymbol{y})^2 \pm \epsilon}\right] \tag{47}$$

$$\leq \sqrt{\mathbb{E}\left[\sum_{i=1}^{n}(1-\eta\tilde{\lambda}_i)^{2t}(\tilde{\boldsymbol{u}}_i^T\boldsymbol{y})^2\right]} \tag{48}$$

$$\leq \sqrt{\sum_{i=1}^{n}\mathbb{E}\left[(1-\eta\tilde{\lambda}_i)^{2t}((\boldsymbol{u}_i\boldsymbol{y})^2 + 2n\sqrt{2}\|E\|/\gamma_0)\right]} \tag{49}$$

$$\leq \sqrt{\sum_{i=1}^{n}(1-\eta\mathbb{E}\left[\tilde{\lambda}_i\right])^{2t}((\boldsymbol{u}_i\boldsymbol{y})^2 + 2n\sqrt{2}\|E\|/\gamma_0)} \tag{50}$$

$$= \sqrt{\sum_{i=1}^{n}\left(1 - \eta\left(\sigma_i^2 + \sigma_i(1-2p_i)\|E\| + \frac{\|E\|^2}{3}\right)\right)^{2t}((\boldsymbol{u}_i\boldsymbol{y})^2 + 2n\sqrt{2}\|E\|/\gamma_0)} \tag{51}$$

### B.3  Convergence analysis for training on the coreset and its augmentation

**Theorem B.2.** *Let $\mathcal{L}_i$ be $\beta$-smooth, $\mathcal{L}$ be $\lambda$-smooth and satisfy the $\alpha$-PL condition, that is for $\alpha > 0$, $\|\nabla\mathcal{L}(\boldsymbol{W}, \boldsymbol{X})\|^2 \geq \alpha\mathcal{L}(\boldsymbol{W}, \boldsymbol{X})$ for all weights $\boldsymbol{W}$. Let $\xi$ upper-bound the normed difference in gradients between the weighted coreset and full dataset. Assume that the network $f(\boldsymbol{W}, \boldsymbol{X})$ is Lipschitz in $\boldsymbol{W}$, $\boldsymbol{X}$ with Lipschitz constant $L$ and $L'$ respectively, and $\bar{L} = \max\{L, L'\}$. Let $G_0$ the gradient over the full dataset at initialization, $\sigma_{\max}$ the maximum Jacobian singular value at initialization. Choosing perturbation bound $\epsilon_0 \leq \frac{1}{\sigma_{\max}\sqrt{\bar{L}n}}$ where $\sigma_{\max}$ is the maximum singular value of the coreset Jacobian and $n$ is the size of the original dataset, running SGD on the coreset and its augmentation using constant step size $\eta = \frac{\alpha}{\lambda\beta}$, we get the following convergence bound:*

$$\mathbb{E}[\|\nabla\mathcal{L}(\boldsymbol{W}^t, \boldsymbol{X}_{c+c_{\text{aug}}})\|] \leq \frac{1}{\sqrt{\alpha}}\left(1 - \frac{\alpha\eta}{2}\right)^{\frac{t}{2}}\left(2G_0 + 2\xi + \mathcal{O}\left(\frac{\bar{L}}{\sigma_{\max}}\right)\right), \tag{52}$$

*where $\boldsymbol{X}_{c+c_{aug}}$ represents the dataset containing the (weighted) coreset and its augmentation.*

*Proof.* As in the proof for Theorem 5.2, we begin with the following inequality

$$\mathbb{E}[\|\nabla\mathcal{L}(\boldsymbol{W}^t, \boldsymbol{X}_{c+c_{\text{aug}}})\|^2] \leq \left(1 - \frac{\alpha\eta}{2}\right)^t \mathcal{L}(\boldsymbol{W}^0, \boldsymbol{X}_{c+c_{\text{aug}}}) \tag{53}$$

$$\leq \frac{1}{\alpha}\left(1 - \frac{\alpha\eta}{2}\right)^t \|\nabla\mathcal{L}(\boldsymbol{W}^0, \boldsymbol{X}_{c+c_{\text{aug}}})\|^2 \tag{54}$$

Thus, we can write

$$\mathbb{E}[\|\nabla\mathcal{L}(\boldsymbol{W}^0, \boldsymbol{X}_{c+c_{\text{aug}}})\|] \tag{55}$$

$$\leq \sqrt{\mathbb{E}[\|\nabla\mathcal{L}(\boldsymbol{W}^t, \boldsymbol{X}_{c+c_{\text{aug}}})\|^2]} \tag{56}$$

$$\leq \frac{1}{\sqrt{\alpha}}\left(1 - \frac{\alpha\eta}{2}\right)^{\frac{t}{2}} \|\nabla\mathcal{L}(\boldsymbol{W}^0, \boldsymbol{X}_{c+c_{\text{aug}}})\| \tag{57}$$

$$\leq \frac{1}{\sqrt{\alpha}}\left(1 - \frac{\alpha\eta}{2}\right)^{\frac{t}{2}} \left(\|\nabla\mathcal{L}(\boldsymbol{W}^0, \boldsymbol{X}_c)\| + \|\nabla\mathcal{L}(\boldsymbol{W}^0, \boldsymbol{X}_{c_{\text{aug}}})\|\right) \tag{58}$$

$$\leq \frac{1}{\sqrt{\alpha}}\left(1 - \frac{\alpha\eta}{2}\right)^{\frac{t}{2}} \left(G_0 + \xi + \|(\mathcal{J}(\boldsymbol{W}^0, \boldsymbol{X}_c) + \boldsymbol{E})(-f(\boldsymbol{W}^0, \boldsymbol{X}_c + \boldsymbol{\epsilon}))\|\right) \tag{59}$$

The rest of the proof is similar to that of Theorem 5.2. $\qquad\square$

## B.4 Lemma for eigenvalues of coreset

The following Lemma characterizes the sum of eigenvalues of the NTK associated with the coreset.

**Lemma B.3.** *Let $\xi$ be an upper bound of the normed difference in gradient of the weighted coreset and the original dataset, i.e. for full data $\boldsymbol{X}$ and its corresponding coreset $\boldsymbol{X}_S$ with weights $\gamma_S$, and respective residuals $\boldsymbol{r}$, $\boldsymbol{r}_S$, we have the bound $\|\mathcal{J}^T(\boldsymbol{W}^t, \boldsymbol{X})\boldsymbol{r}^t - \gamma_S\mathcal{J}^T(\boldsymbol{W}^t, \boldsymbol{X}_S)\boldsymbol{r}_S^t\| \leq \xi$. Let $\{\lambda_i\}_{i=1}^k$ be the eigenvalues of the NTK associated with the coreset. Then we have that*

$$\sqrt{\sum_{i=1}^k \lambda_i} \geq \frac{|\|\mathcal{J}^T(\boldsymbol{W}^t, \boldsymbol{X})\boldsymbol{r}^t\| - \xi|}{\|\boldsymbol{r}_S^t\|}.$$

*Proof.* Let singular values of coreset Jacobian be $\sigma_i$. Let $\mathcal{J}^T(\boldsymbol{W}^t, \boldsymbol{X})\boldsymbol{r}^t = \gamma_S\mathcal{J}^T(\boldsymbol{W}^t, \boldsymbol{X}_S)\boldsymbol{r}_S^t + \xi_S$ where $\|\xi_S\| \leq \xi$.

Taking Frobenius norm, we get

$$\|\gamma_S\mathcal{J}^T(\boldsymbol{W}^t, \boldsymbol{X}_S)\boldsymbol{r}_S^t\| = \|\mathcal{J}^T(\boldsymbol{W}^t, \boldsymbol{X})\boldsymbol{r}^t - \xi_S\| \tag{60}$$

$$\Rightarrow \|\gamma_S\mathcal{J}^T(\boldsymbol{W}^t, \boldsymbol{X}_S)\|\|\boldsymbol{r}_S^t\| \geq \|\mathcal{J}^T(\boldsymbol{W}^t, \boldsymbol{X})\boldsymbol{r}^t - \xi_S\| \tag{61}$$

$$\Rightarrow \|\gamma_S\mathcal{J}^T(\boldsymbol{W}^t, \boldsymbol{X}_S)\| \geq \frac{\|\mathcal{J}^T(\boldsymbol{W}^t, \boldsymbol{X})\boldsymbol{r}^t - \xi_S\|}{\|\boldsymbol{r}_S^t\|} \tag{62}$$

$$\Rightarrow \sqrt{\sum_{i=1}^s \sigma_i^2} \geq \frac{\|\mathcal{J}^T(\boldsymbol{W}^t, \boldsymbol{X})\boldsymbol{r}^t - \xi_S\|}{\|\boldsymbol{r}_S^t\|} \tag{63}$$

$$\Rightarrow \sqrt{\sum_{i=1}^s \lambda_i} \geq \frac{\|\mathcal{J}^T(\boldsymbol{W}^t, \boldsymbol{X})\boldsymbol{r}^t - \xi_S\|}{\|\boldsymbol{r}_S^t\|} \tag{64}$$

$$\Rightarrow \sqrt{\sum_{i=1}^s \lambda_i} \geq \frac{|\|\mathcal{J}^T(\boldsymbol{W}^t, \boldsymbol{X})\boldsymbol{r}^t\| - \xi|}{\|\boldsymbol{r}_S^t\|} \quad \text{by reverse triangle inequality} \tag{65}$$

$$\square$$

We can make the following observations: For overparameterized networks, with bounded activation functions and labels, e.g. softmax and one-hot encoding, the norm of the residual vector is bounded,

and the gradient norm is likely to be much larger than residual, especially when dimension of gradient is large. In this case, the Jacobian matrix associated with small weighted coresets found by solving Eq. (9), have large singular values.

## B.5 Augmentation as Linear Transformation: Linear Model Analysis

We introduce a simplified linear model to extend our theoretical analysis to augmentations modelled as linear transformation matrices $F$ applied to the original training data. These augmentations are also originally studied by [7]. In this section, we specifically study the effect of these augmentations using a linear model when applied to coresets.

**Lemma B.4** (Augmented coreset gradient bounds: Linear). *Let $f$ be a simple linear model with weights $\boldsymbol{W} \in \mathbb{R}^{d \times C}$ where $f(\boldsymbol{W}, \boldsymbol{x}_i) = \boldsymbol{W}^T \boldsymbol{x}_i$, trained on mean squared loss function $\mathcal{L}$. Let $F \in \mathbb{R}^{d \times d}$ be a common linear augmentation matrix with norm $\|F\|$ with augmentation $\boldsymbol{x}_i^{\mathrm{aug}}$ given by $F\boldsymbol{x}_i$. Let coreset be of size $k$ and full dataset be of size $n$. Further assume that the predicted label of $\boldsymbol{x}_i$ and its augmentation $F\boldsymbol{x}_i$ are sufficiently close, i.e. there exists $\omega$ such that $\boldsymbol{W}^T(F\boldsymbol{x}_i) = \boldsymbol{W}^T \boldsymbol{x}_i + z_i$, $\|z_i\| \leq \omega$ $\forall i$. Let $\xi$ upper-bound the normed difference in gradients between the weighted coreset and full dataset. Then, the normed difference between the gradient of the augmented full data and augmented coreset is given by*

$$\|\sum_{i \in V} \nabla\mathcal{L}(\boldsymbol{W}, \boldsymbol{x}_i^{\mathrm{aug}}) - \sum_{j=1}^{k} \gamma_{s_j} \nabla\mathcal{L}(\boldsymbol{W}, \boldsymbol{x}_{s_j}^{\mathrm{aug}})\| \leq \|F\|(\xi + \sqrt{d}n\omega)$$

*for some (small) constant $\xi$.*

*Proof.* By our assumption, we can begin with,

$$\|\sum_{i \in V} \nabla\mathcal{L}(\boldsymbol{W}, \boldsymbol{x}_i) - \sum_{j=1}^{k} \gamma_{s_j} \nabla\mathcal{L}(\boldsymbol{W}, \boldsymbol{x}_{s_j})\| \leq \xi \tag{66}$$

Furthermore, by [6], we know that sum of the coreset weights $\gamma_{s_j}$ is given by $\sum_{j=1}^{k=1} \gamma_{s_j} \leq n$. Hence,

$$\|\sum_{i \in V} \nabla\mathcal{L}(\boldsymbol{W}, \boldsymbol{x}_i^{\mathrm{aug}}) - \sum_{j=1}^{k} \gamma_{s_j} \nabla\mathcal{L}(\boldsymbol{W}, \boldsymbol{x}_{s_j}^{\mathrm{aug}})\| \tag{67}$$

$$= \|\sum_{i \in V} (\mathcal{J}(\boldsymbol{W}, \boldsymbol{x}_i^{\mathrm{aug}}))^T [\boldsymbol{W}^T(F\boldsymbol{x}_i) - y_i] - \sum_{j=1}^{k} \gamma_{s_j} (\mathcal{J}(\boldsymbol{W}, \boldsymbol{x}_{s_j}^{\mathrm{aug}}))^T [\boldsymbol{W}^T(F\boldsymbol{x}_{s_j}) - y_{s_j}]\| \tag{68}$$

$$= \|\sum_{i \in V} F\boldsymbol{x}_i [\boldsymbol{W}^T(F\boldsymbol{x}_i) - y_i] - \sum_{j=1}^{k} \gamma_{s_j} F\boldsymbol{x}_{s_j} [\boldsymbol{W}^T(F\boldsymbol{x}_{s_j}) - y_{s_j}]\| \tag{69}$$

$$= \|F \sum_{i \in V} \boldsymbol{x}_i (\boldsymbol{W}^T \boldsymbol{x}_i - y_i) - F \sum_{j=1}^{k} \gamma_{s_j} \boldsymbol{x}_{s_j} (\boldsymbol{W}^T \boldsymbol{x}_{s_j} + z_i - y_{s_j})\| \tag{70}$$

$$= \|F \sum_{i \in V} \nabla L(\boldsymbol{W}, \boldsymbol{x}_i) - F \sum_{j=1}^{k} \gamma_{s_j} \nabla L(\boldsymbol{W}, \boldsymbol{x}_{s_j}) - F \sum_{j=1}^{k} \gamma_{s_j} \boldsymbol{x}_{s_j} z_{s_j}\| \tag{71}$$

$$\leq \|F\| \|\sum_{i \in V} \nabla L(\boldsymbol{W}, \boldsymbol{x}_i) - \sum_{j=1}^{k} \gamma_{s_j} \nabla L(\boldsymbol{W}, \boldsymbol{x}_{s_j})\| + \|F\| \|\sum_{j=1}^{k} \gamma_{s_j} \boldsymbol{x}_{s_j} z_{s_j}\| \tag{72}$$

$$\leq \|F\|\xi + \sqrt{d}\|F\|n\omega \tag{73}$$

$$= \|F\|(\xi + \sqrt{d}n\omega) \tag{74}$$

$$\square$$

**Corollary B.5.** *In the simplified linear case above, the difference in gradients of the full training data with its augmentations ($\nabla\mathcal{L}(\boldsymbol{W}, \boldsymbol{X}_{f+aug})$) and gradients of the coreset with its augmentations ($\nabla\mathcal{L}(\boldsymbol{W}, \boldsymbol{X}_{c+c_{aug}})$) can be bounded by*

$$\|\nabla\mathcal{L}(\boldsymbol{W}, \boldsymbol{X}_{f+aug}) - \nabla\mathcal{L}(\boldsymbol{W}, \boldsymbol{X}_{c+c_{aug}})\| \leq (\|F\|+1)\xi + \sqrt{d}\|F\|n\omega$$

*Proof.* Applying Eq. (66) and Lemma B.4, we obtain

$$\|\nabla\mathcal{L}(\boldsymbol{W}, \boldsymbol{X}_{f+aug}) - \nabla\mathcal{L}(\boldsymbol{W}, \boldsymbol{X}_{c+c_{aug}})\| \tag{75}$$
$$= \|(\nabla\mathcal{L}(\boldsymbol{W}, \boldsymbol{X}_f) + \nabla\mathcal{L}(\boldsymbol{W}, \boldsymbol{X}_{aug})) - (\nabla\mathcal{L}(\boldsymbol{W}, \boldsymbol{X}_c) + \nabla\mathcal{L}(\boldsymbol{W}, \boldsymbol{X}_{c_{aug}}))\| \tag{76}$$
$$= \|(\nabla\mathcal{L}(\boldsymbol{W}, \boldsymbol{X}_f) - \nabla\mathcal{L}(\boldsymbol{W}, \boldsymbol{X}_c)) + (\nabla\mathcal{L}(\boldsymbol{W}, \boldsymbol{X}_{aug}) - \nabla\mathcal{L}(\boldsymbol{W}, \boldsymbol{X}_{c_{aug}}))\| \tag{77}$$
$$\leq \|(\nabla\mathcal{L}(\boldsymbol{W}, \boldsymbol{X}_f) - \nabla\mathcal{L}(\boldsymbol{W}, \boldsymbol{X}_c))\| + \|(\nabla\mathcal{L}(\boldsymbol{W}, \boldsymbol{X}_{aug}) - \nabla\mathcal{L}(\boldsymbol{W}, \boldsymbol{X}_{c_{aug}}))\| \tag{78}$$
$$\leq \xi + \|F\|(\xi + \sqrt{d}n\omega) \tag{79}$$
$$= (\|F\|+1)\xi + \sqrt{d}\|F\|n\omega \tag{80}$$

$\square$

**Theorem B.6** (Convergence of linear model). *Let $f$ be a linear model with weights $\boldsymbol{W}$ and augmentation be represented by the common linear transformation $F$. Let $\mathcal{L}_i$ be $\beta$-smooth, $\mathcal{L}$ be $\lambda$-smooth and satisfy the $\alpha$-PL condition, that is for $\alpha > 0$, $\|\nabla\mathcal{L}(\boldsymbol{W}, \boldsymbol{X})\|^2 \geq \alpha\mathcal{L}(\boldsymbol{W}, \boldsymbol{X})$ for all weights $\boldsymbol{W}$. Let $\xi$ upper-bound the normed difference in gradients between the weighted coreset and full dataset and $\omega$ bound $\boldsymbol{W}^T(F\boldsymbol{x}_i) = \boldsymbol{W}^T\boldsymbol{x}_i + z_i$, $\|z_i\| \leq \omega\ \forall i$. Let $G_0'$ be the gradient over the full dataset and its augmentations at initialization. Then, running SGD on the size $k$ coreset with its augmentation using constant step size $\eta = \frac{\alpha}{\lambda\beta}$, we get the following convergence bound:*

$$\mathbb{E}[\|\nabla\mathcal{L}(\boldsymbol{W}^t, \boldsymbol{X}_{c+c_{\text{aug}}})\|] \leq \frac{1}{\sqrt{\alpha}}\left(1 - \frac{\alpha\eta}{2}\right)^{\frac{t}{2}}\left(G_0' + (\|F\|+1)\xi + \sqrt{d}\|F\|n\omega\right)$$

*Proof.* From [2], we have

$$\mathbb{E}[\|\nabla\mathcal{L}(\boldsymbol{W}^t, \boldsymbol{X}_{c+c_{\text{aug}}})\|^2] \leq \left(1 - \frac{\alpha\eta}{2}\right)^t \mathcal{L}(\boldsymbol{W}^0, \boldsymbol{X}_{c+c_{\text{aug}}}) \tag{81}$$
$$\leq \frac{1}{\alpha}\left(1 - \frac{\alpha\eta}{2}\right)^t \|\nabla\mathcal{L}(\boldsymbol{W}^0, \boldsymbol{X}_{c+c_{\text{aug}}})\|^2 \tag{82}$$
$$\tag{83}$$

Using Jensen's inequality, we have

$$\mathbb{E}[\|\nabla\mathcal{L}(\boldsymbol{W}^t, \boldsymbol{X}_{c+c_{\text{aug}}})\|] \tag{84}$$
$$\leq \sqrt{\mathbb{E}[\|\nabla\mathcal{L}(\boldsymbol{W}^t, \boldsymbol{X}_{c+c_{\text{aug}}})\|^2]} \tag{85}$$
$$\leq \frac{1}{\sqrt{\alpha}}\left(1 - \frac{\alpha\eta}{2}\right)^{\frac{t}{2}}\|\nabla\mathcal{L}(\boldsymbol{W}^0, \boldsymbol{X}_{c+c_{\text{aug}}})\| \tag{86}$$
$$\leq \frac{1}{\sqrt{\alpha}}\left(1 - \frac{\alpha\eta}{2}\right)^{\frac{t}{2}}\left(G_0' + (\|F\|+1)\xi + \sqrt{d}\|F\|n\omega\right) \tag{87}$$

where the last inequality follows from applying Corollary B.5. $\square$

## C Singular spectrum analysis

### C.1 Experiment details

We generate singular spectrum plots for both MNIST and CIFAR10 datasets in Figure 1. Due to the computational infeasbility of computing the network Jacobian for the full datasets in deep network settings, we instead construct and use a reduced version of these datasets by uniformly select 900 images from the first 3 classes. For our experiments on MNIST, we pretrain a MLP model with 1

hidden layer for 15 epochs. For our experiments on CIFAR10, we pretrain a ResNet20 model for 15 epochs. We then compute the singular spectrums for augmented and non-augmented data based on these pretrained networks.

Since it is difficult to perform a one-to-one matching of singular values produced from augmented and non-augmented datasets, we instead bin our singular values into 30 separate and uniformly distributed bins each containing the same number of singular values. To measure perturbation to singular values resulted from augmentation, we compute the mean difference between each bin. On the other hand, to measure perturbation to singular vectors, we compute mean subspace angle between the singular subspace spanned by singular vectors in each bin.

### C.2 Real-world strong augmentations

We study the effects of real-world, unbounded augmentations on the singular spectrum of the network Jacobian. In particular, in additional to the plots in the main paper, we show the effect of strong augmentations through (1) random rotation (up to 30◦ and AutoAugment [3] for MNIST and (2) random horizontal flips/random crops and AutoAugment for CIFAR10. The policies implemented by AutoAugment include translations, shearing, as well as contrast and brightness transforms. We study the effects of these augmentations on the singular spectrum in Figure 1. Despite these augmentations being unbounded transformations, we note that the results of our theory still holds. In particular, it can be observed that data augmentation increases smaller singular values relatively more with a higher probability. On the other hand, data augmentation affects the prominent singular vectors of the Jacobian to a smaller extent, and preserves the prominent directions. As such, our argument empirically extends to real-world, unbounded label-invariant transformations characteristic of strong augmentations.

## D Experiment Setup and Additional Experiments

### D.1 Experiment setup

For all experiments, we train using SGD with 0.9 momentum and learning rate decay. For experiments on CIFAR10 and variants/ResNet20, we train for 200 epochs, for Caltech256 (ImageNet pretrained)/ ResNet18, we trained for 40 epochs starting at learning rate $0.001$ and batch size 64. We also report results for Caltech256 without ImageNet pretraining in Sec. D.8, where we train for 400 epochs to ensure convergence with a starting learning rate of $0.05$ and batch size 64. For experiments on ImageNet/ResNet50 and TinyImageNet/ResNet50, we use the standard 90 epoch learning schedule starting at learning rate of $0.1$ and batch size 64.

**Data and augmentation.** We apply our method to training ResNet20 and Wide-ResNet-28-10 on CIFAR10, and ResNet32 on CIFAR10-IMB (Long-Tailed CIFAR10 with Imbalance factor of 100 following [5]) and SVHN datasets. We train Caltech256 [4] on ImageNet-pretrained ResNet18, and include experiments with random initialization in Appendix D. TinyImageNet and ImageNet are trained on with ResNet50. We use [7] for CIFAR10/SVHN, and AutoAugment [3] for Caltech256, TinyImageNet, and ImageNet as the strong augmentation method. Note that we append strong augmentations rather than apply them in-place, which we show to be more effective in Appendix D. All results are averaged over 5 runs using an Nvidia A40 GPU.

### D.2 Experiment setup

For all experiments, we train using SGD with 0.9 momentum and learning rate decay. We also set weight decay as For experiments on CIFAR10 and variants/ResNet20, we train for 200 epochs, for Caltech256 (ImageNet pretrained)/ ResNet18, we trained for 40 epochs starting at learning rate $0.001$ and batch size 64. We also report results for Caltech256 without ImageNet pretraining in Sec. D.8, where we train for 400 epochs to ensure convergence with a starting learning rate of $0.05$ and batch size 64. For experiments on ImageNet/ResNet50 and TinyImageNet/ResNet50, we use the standard 90 epoch learning schedule starting at learning rate of $0.1$ and batch size 64.

**Data and augmentation.** We apply our method to training ResNet20 and Wide-ResNet-28-10 on CIFAR10, and ResNet32 on CIFAR10-IMB (Long-Tailed CIFAR10 with Imbalance factor of 100 following [5]) and SVHN datasets. We train Caltech256 [4] on ImageNet-pretrained ResNet18, and

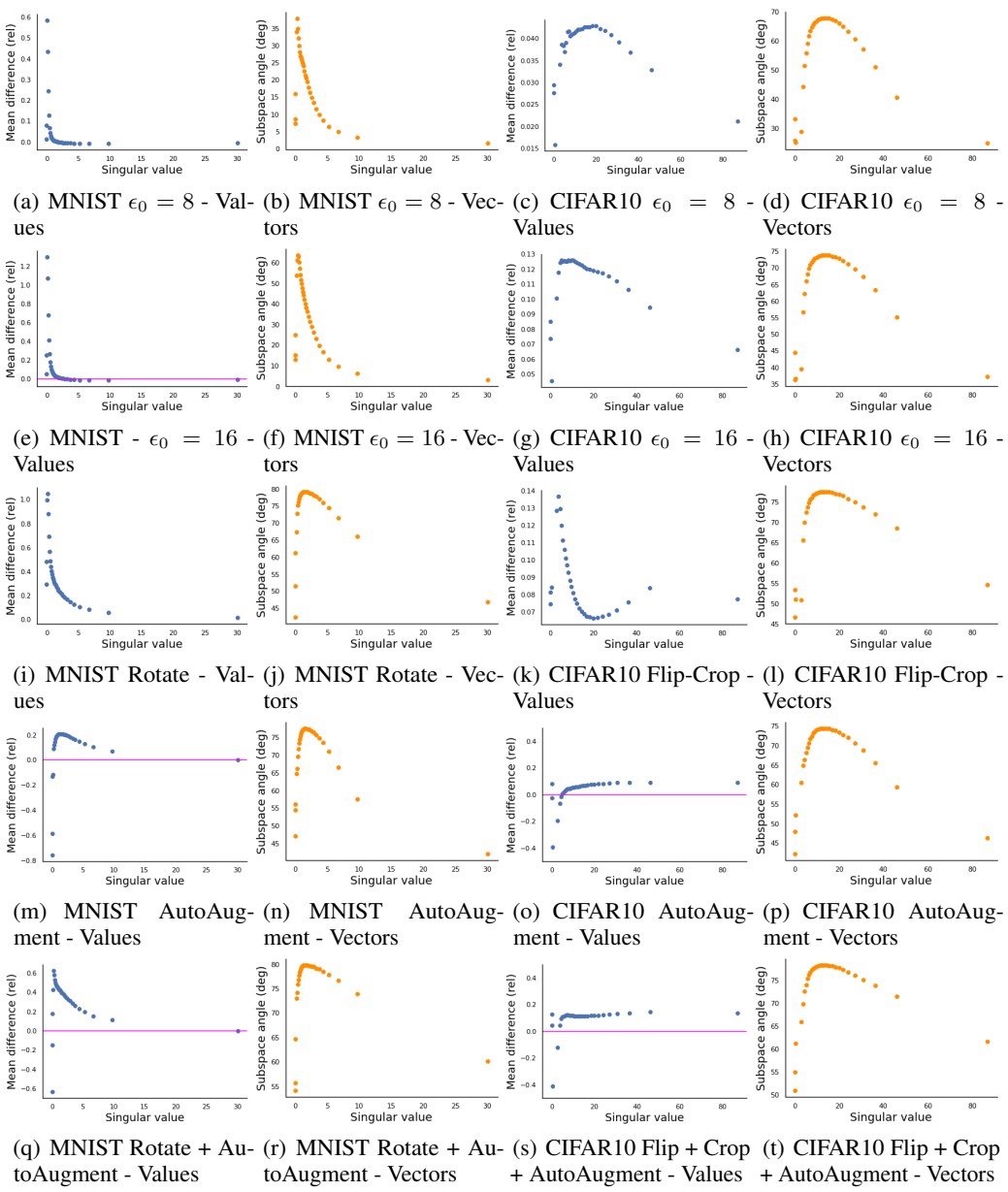

(a) MNIST $\epsilon_0 = 8$ - Values
(b) MNIST $\epsilon_0 = 8$ - Vectors
(c) CIFAR10 $\epsilon_0 = 8$ - Values
(d) CIFAR10 $\epsilon_0 = 8$ - Vectors

(e) MNIST - $\epsilon_0 = 16$ - Values
(f) MNIST $\epsilon_0 = 16$ - Vectors
(g) CIFAR10 $\epsilon_0 = 16$ - Values
(h) CIFAR10 $\epsilon_0 = 16$ - Vectors

(i) MNIST Rotate - Values
(j) MNIST Rotate - Vectors
(k) CIFAR10 Flip-Crop - Values
(l) CIFAR10 Flip-Crop - Vectors

(m) MNIST AutoAugment - Values
(n) MNIST AutoAugment - Vectors
(o) CIFAR10 AutoAugment - Values
(p) CIFAR10 AutoAugment - Vectors

(q) MNIST Rotate + AutoAugment - Values
(r) MNIST Rotate + AutoAugment - Vectors
(s) CIFAR10 Flip + Crop + AutoAugment - Values
(t) CIFAR10 Flip + Crop + AutoAugment - Vectors

Figure 1: Difference in mean singular values (Cols 1 & 3) between augmented and non-augmented data and mean angular difference (Cols 2 & 4) between subspaces spanned by singular vectors for augmented and non-augmented data.

include experiments with random initialization in Appendix D. TinyImageNet and ImageNet are trained on with ResNet50. We use [7] for CIFAR10/SVHN, and AutoAugment [3] for Caltech256, TinyImageNet, and ImageNet as the strong augmentation method. Note that we append strong augmentations rather than apply them in-place, which we show to be more effective in Appendix D. All results are averaged over 5 runs using an Nvidia A40 GPU.

### D.3 Full Results for Table 4

This section contains full experiment results including standard deviations and the full augmentation benchmark for Table 4. Augmenting coresets of size 10% achieves 51% of the improvement obtained from augmentation of the full data and further enjoys a 6x speedup in training time on CIFAR10.

This speedup becomes more significant when using strong augmentation techniques which are mostly computationally demanding, especially when applied to the entire dataset.

Table 1: Supplementary table for Table 4 - Test accuracy on CIFAR10 + ResNet20, SVHN + ResNet32, CIFAR10-Imbalanced + ResNet32 including standard deviation errors and full dataset augmentation accuracy.

| Method | Size | CIFAR10 | CIFAR10-IMB | SVHN |
|---|---|---|---|---|
| None | 0% | $89.46 \pm 0.17\%$ | $87.08 \pm 0.50\%$ | $95.676 \pm 0.108\%$ |
| Random | 5% | $90.34 \pm 0.18\%$ | $88.48 \pm 0.25\%$ | $95.760 \pm 0.084\%$ |
| | 10% | $91.07 \pm 0.13\%$ | $89.52 \pm 0.15\%$ | $96.187 \pm 0.112\%$ |
| | 30% | $92.11 \pm 0.12\%$ | $91.11 \pm 0.18\%$ | $96.569 \pm 0.073\%$ |
| Max-Loss | 5% | $90.79 \pm 0.19\%$ | $88.77 \pm 0.35\%$ | $\mathbf{96.165 \pm 0.108}\%$ |
| | 10% | $91.39 \pm 0.08\%$ | $89.22 \pm 0.48\%$ | $\mathbf{96.370 \pm 0.076}\%$ |
| | 30% | $92.43 \pm 0.07\%$ | $91.11 \pm 0.25\%$ | $96.735 \pm 0.068\%$ |
| Coreset | 5% | $\mathbf{90.87 \pm 0.05}\%$ | $\mathbf{89.10 \pm 0.41}\%$ | $96.121 \pm 0.055\%$ |
| | 10% | $\mathbf{91.54 \pm 0.19}\%$ | $\mathbf{89.75 \pm 0.52}\%$ | $96.354 \pm 0.091\%$ |
| | 30% | $\mathbf{92.49 \pm 0.15}\%$ | $\mathbf{91.12 \pm 0.26}\%$ | $\mathbf{96.791 \pm 0.051}\%$ |
| All | 100% | $93.50 \pm 0.25\%$ | $92.48 \pm 0.34\%$ | $97.068 \pm 0.030\%$ |

## D.4 Supplementary results for Table 1

Table 2: Supplementary results for Tab. 1. Training ResNet20 (R20) and WideResnet-28-10 (W2810) on CIFAR10 (C10) using small subsets, and ResNet18 (R18) on Caltech256 (Cal).

| Model/Dataset | Subset | Random | | Ours | |
|---|---|---|---|---|---|
| | | Weak Aug. | Strong Aug. | Weak Aug. | Strong Aug. |
| C10/R20 | 0.1% (5) | $31.7 \pm 3.2$ | $33.5 \pm 2.7$ | $29.6 \pm 3.8$ | $\mathbf{37.8 \pm 4.5}$ |
| | 0.2% (10) | $35.9 \pm 2.1$ | $42.7 \pm 3.9$ | $33.6 \pm 3.2$ | $\mathbf{45.1 \pm 2.3}$ |
| | 0.5% (25) | $51.1 \pm 2.3$ | $58.7 \pm 1.3$ | $55.8 \pm 3.1$ | $\mathbf{63.9 \pm 2.1}$ |
| | 1% (50) | $66.2 \pm 1.0$ | $74.4 \pm 0.8$ | $65.9 \pm 4.0$ | $\mathbf{74.7 \pm 1.1}$ |
| C10/W2810 | 1% (50) | $61.3 \pm 2.4$ | $57.7 \pm 0.8$ | $59.9 \pm 2.4$ | $\mathbf{62.1 \pm 3.1}$ |
| Cal/R18 | 5% (3) | $24.8 \pm 1.5$ | $41.5 \pm 0.5$ | $33.8 \pm 1.7$ | $\mathbf{52.7 \pm 1.2}$ |
| | 10% (6) | $49.5 \pm 0.6$ | $61.8 \pm 0.8$ | $55.7 \pm 0.3$ | $\mathbf{65.4 \pm 0.3}$ |
| | 20% (12) | $66.6 \pm 0.2$ | $72.5 \pm 0.1$ | $67.5 \pm 0.3$ | $\mathbf{73.1 \pm 0.1}$ |
| | 30% (18) | $72.0 \pm 0.1$ | $75.7 \pm 0.2$ | $71.9 \pm 0.2$ | $\mathbf{76.3 \pm 0.2}$ |
| | 40% (24) | $74.6 \pm 0.3$ | $77.6 \pm 0.4$ | $74.2 \pm 0.4$ | $\mathbf{77.7 \pm 0.5}$ |
| | 50% (30) | $76.1 \pm 0.5$ | $78.5 \pm 0.3$ | $76.1 \pm 0.1$ | $\mathbf{78.9 \pm 0.2}$ |

## D.5 Training dynamics vs generalization

Figure 2 demonstrates the relationship between training loss and validation accuracy resulted from data augmentation. While training loss of augmented datasets do not decrease as quickly as non-augmented datasets, generalization performance (shown by val. acc.) improves.

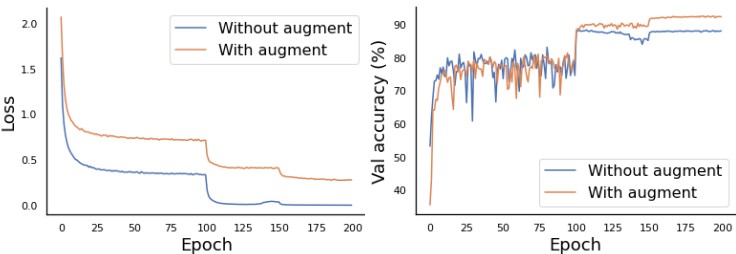

Figure 2: Training loss vs validation accuracy of CIFAR10/ResNet20 using AutoAugment.

## D.6 Augmentations applied through appending vs in-place

Our experiments on Caltech256/ResNet18/AutoAugment ($R$=5) show that even for cheaper strong augmentation methods (AutoAugment), while in-place augmentation may decrease the performance, appending Random (R) and Coresets (C) augmentations (Append) outperforms in-place augmentation of 2x data points (In-place 2x) for various subset sizes.

Table 3: Caltech256/AutoAugment in-place vs. appending for Caltech256.

|      | No Aug. | In-place | In-place (2x) | Append |
|------|---------|----------|---------------|--------|
| C5%  | 33.8%   | 26.4%    | 48.2%         | **52.7%** |
| R5%  | 24.8%   | 17.4%    | 40.2%         | **41.5%** |
| C10% | 55.7%   | 48.2%    | 62.8%         | **65.4%** |
| R10% | 50.6%   | 40.2%    | **62.0%**     | 61.8%  |
| C30% | 71.9%   | 68.8%    | 74.9%         | **76.3%** |
| R30% | 72.0%   | 68.7%    | 75.1%         | **75.7%** |

## D.7 Speed-up measurements

We measure the improvement in training time in the case of training on full data and augmenting subsets of various sizes. While our method yields similar or slightly lower speed-up to the max-loss policy and random approach respectively, our resulting accuracy outperforms these two approaches. We show this in Fig. D.7. For example, for SVHN/Resnet32 using 30% coresets, we sacrifice 11% of the speed-up to obtain an additional 24.8% of the gain in accuracy from full data augmentation when compared to a random subset of the same size. We show the speed-up obtained for our method and various subset sizes in Tab. 4, and provide wall-clock times for our method in Tab. D.7.

Table 4: Speedup on CIFAR10 + ResNet20 (C10/R20), SVHN + ResNet32 (SVHN/R32).

| Dataset | Full Aug. | Ours | | | | | | Max loss. | Random. |
|---------|-----------|------|------|------|------|------|------|-----------|---------|
|         | 100%      | 5%   | 10%  | 15%  | 20%  | 25%  | 30%  | 30%       | 30%     |
| C10 / R20 | 1x | 7.93x | 6.31x | 4.46x | 4.27x | 3.41x | 3.43x | 3.48x | 4.03x |
| SVHN / R32 | 1x | 5.35x | 3.93x | 3.40x | 2.80x | 2.49x | 2.18x | 2.21x | 2.43x |

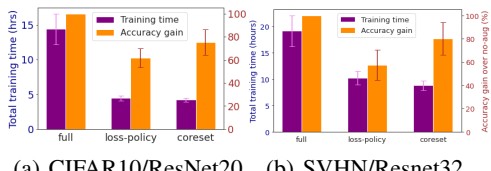

(a) CIFAR10/ResNet20   (b) SVHN/Resnet32

Figure 3: Speedup/Accuracy of augmenting 30% coresets compared to original max-loss policy for (a) ResNet20 trained on CIFAR10 and (b) ResNet32 trained on SVHN.

Table 5: Wall-clock times to find various sized coresets from *all classes* of Caltech256 and TinyImagene at 1 epoch. Note, training ResNet20/CIFAR10 with [7] takes 14.4 hrs. In practice, coresets can be found in parallel ($p$ threads) from different classes, and selection happens every $R=5-15$ epochs. Hence, the numbers divide by $p \times R$.

| | Caltech256 | | | TinyImageNet | |
|---|---|---|---|---|---|
| 10% | 30% | 50% | 10% | 30% | 50% |
| 10.50s | 10.52s | 10.53s | 7.85s | 8.09s | 8.17s |

## D.8  End to end training on Caltech256

As Caltech256 contains many classes and higher resolution images, training on smaller subset without pretraining has a low accuracy. Thus, many works (e.g. Achille et al., 2020) finetune from ImageNet pretrained initialization. However, we show that our results still hold even when training form scratch. We demonstrate our results in Tab. 6, where we train Caltech256 on ResNet50 without pretraining for 400 epochs, and with $R = 40$, where our method consistently outperfoms random subsets for multiple subset sizes (5%, 10%, 30%, 50%).

Table 6: Caltech256 (w/o pretraining) /ResNet50, 400 epochs, $R = 40$

| | Random | | | | Ours | | |
|---|---|---|---|---|---|---|---|
| 5% | 10% | 30% | 50% | 5% | 10% | 30% | 50% |
| 17.26 | 35.38 | 58.2 | 64.67 | **20.58** | **38.20** | **60.30** | **65.17** |

## D.9  Training on full data and augmenting small subsets re-selected every epoch

We apply our proposed method to select a new subset for augmentation every epoch (i.e. using $R = 1$) and compare our results with other approaches using accuracy and percentage of data not selected (NS). We see that while the max-loss policy selects a small fraction of data points over and over and random uniformly selects all the data points, our approach successfully finds the smallest subset of data points that are the most crucial for data augmentation. Hence, it can achieve a superior accuracy than max-loss policy, while augmenting only slightly more examples. This confirms the data-efficiency of our approach. This is especially evident when using coresets of size 0.2%. Furthermore, despite the random baseline using a significantly larger percentage of data, it is outperformed by our approach in both data-efficiency and accuracy. We emphasize that results in this table is different from that of Table 1, as default augmentations on the full training data are performed once every $R = 1$ epochs instead of every $R = 20$ epochs. Since selecting subsets at every epoch can be computationally expensive, we only perform these experiments on small coresets and hence still enjoy good speedups compared to full data augmentation. This shows that our approach is still effective at very small subset sizes, hence can be computationally efficient even when subsets are re-selected every epoch.

Table 7: Training on full data and selecting a new subset for augmentation every epoch ($R = 1$).

| Subset | Random | | Max-loss Policy | | Ours | |
|---|---|---|---|---|---|---|
| | Acc | NS (%) | Acc | NS (%) | Acc | NS (%) |
| 0% | $91.96 \pm 0.12$ | – | $91.96 \pm 0.12$ | – | $91.96 \pm 0.12$ | – |
| 0.2% | $92.22 \pm 0.22$ | $67.03 \pm 0.04$ | $91.94 \pm 0.12$ | $86.70 \pm 0.15$ | $\mathbf{92.26} \pm 0.13$ | $79.19 \pm 1.10$ |
| 0.5% | $92.06 \pm 0.17$ | $36.70 \pm 0.18$ | $92.20 \pm 0.13$ | $76.80 \pm 0.31$ | $\mathbf{92.27} \pm 0.08$ | $63.23 \pm 0.35$ |

## D.10  Additional visualizations for training on coresets and its augmentations - Measuring training dynamics over time

We include additional visualizations in Figure 4 for training on coresets and its augmentations as supplementary plots to Figure 7(c) and Table 1. We plot metrics obtained during each point (epoch)

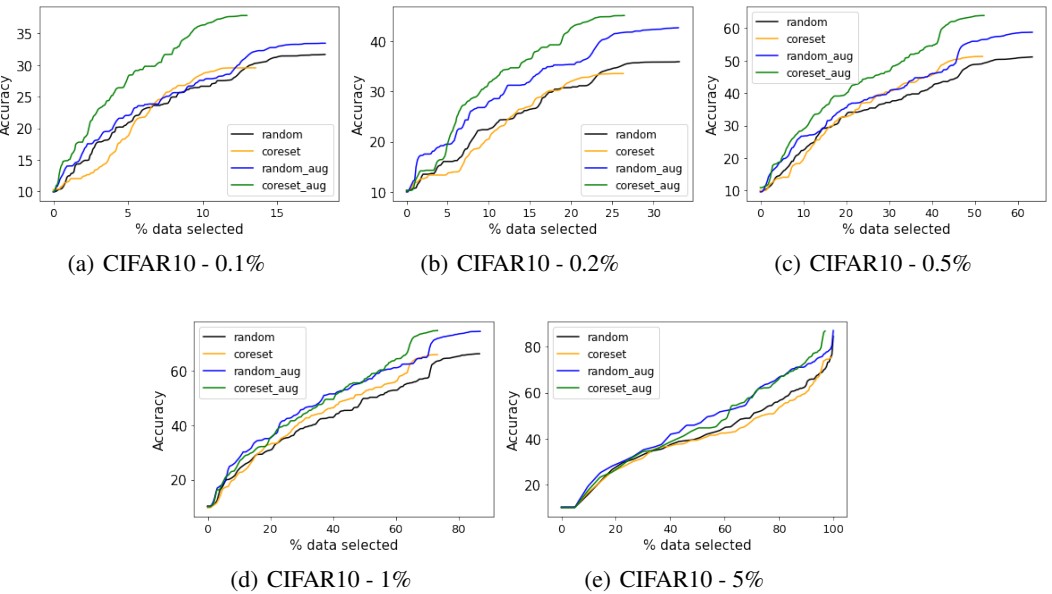

(a) CIFAR10 - 0.1%  (b) CIFAR10 - 0.2%  (c) CIFAR10 - 0.5%

(d) CIFAR10 - 1%  (e) CIFAR10 - 5%

Figure 4: Supplementary plots for Figure 7(c): Training on coreset and its augmentation compared to random baseline, measured using test accuracy against percentage of data used on CIFAR10 dataset across various subset sizes. Accuracy and percentage of data used are measured at every epoch and averaged over 5 runs.

of the training process based on percentage of data selected/used and test accuracy achieved. All metrics are averaged over 5 runs and obtained using $R = 1$. These plots demonstrate that coreset augmentation approaches outperform random augmentation baselines throughout the training process. Furthermore, they show that augmentation of coresets result in a larger increase in test accuracy compared to augmentation of randomly selected training examples, especially for small subset sizes.

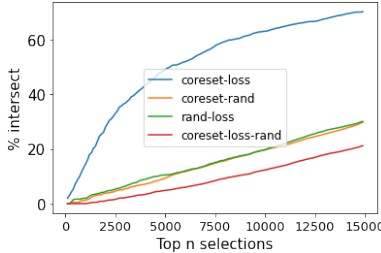

Figure 5: Intersection between max-loss and coresets in the top $N$ points selected aggregated across the entire training process. Here, we show the increasing overlap between max-loss and coreset points as $N$ grows.

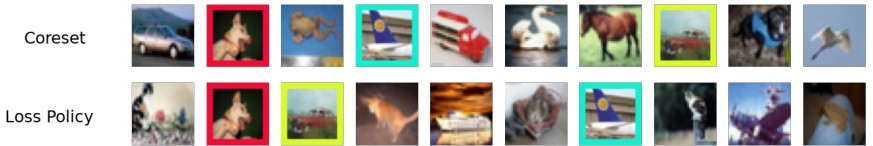

Figure 6: Qualitative evaluation of coreset and max-loss points.

## D.11   Intersection of max-loss policy and coresets

Figure 7($a$) depicts the increase in intersection between max-loss subsets and coresets over time. In addition, we also aggregate $30\%$ subsets selected every $R = 20$ epochs using both approaches over the entire training process to compute intersection between the top $N$ selected data points. Our plots

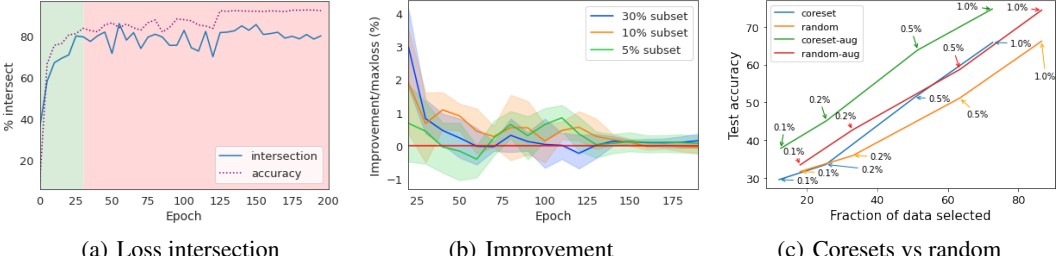

Figure 7: Training ResNet20 on full data and augmented coresets extracted from CIFAR10. (a) Intersection between elements of coresets of size 30% and maximum loss subsets of the same size. The intersection increases after the initial phase of training, (b) Accuracy improvement for training on full data and augmented coresets over training on full data and max-loss augmentation. (c) Accuracy vs. fraction of data selected for augmentation during training Resnet20 on CIFAR10.

in Figure 5 suggest that a similar pattern holds in this setting. We also qualitatively visualize this in Figure 6.