# OpenReview forum: "Data-Efficient Augmentation for Training Neural Networks"
_NeurIPS.cc/2022/Conference — NeurIPS 2022 Accept_

### Official Review · Reviewer_xtob · 2022-07-09

**Rating:** 6
**Confidence:** 4
**Soundness:** 2 fair
**Presentation:** 3 good
**Contribution:** 2 fair

**Summary:**

This paper provides analysis to show that the effect of data augmentations can be explained by looking at the singular values of the network jacobian. Based on this, an algorithm is proposed to find a subset of the data points on which data augmentations can be done to best approximate the effect of doing augmentations on the whole dataset. Experiments show that the proposed algorithm works better than selecting the subset randomly or according to max-loss.

**Questions:**

Please see Weaknesses

**Limitations:**

The authors included some discussions about the limitations.

**Strengths And Weaknesses:**

Strengths:
1. I like the analysis part which focuses on the effect of data augmentation on the singular values of the network jacobian, which is a novel observation and motivates the proposed selection algorithm.
2. Extensive experiments show that the proposed selection algorithm works better than the random selection and the selection according to max-loss.

Weaknesses:
I have two major concerns:
1. The problem itself is not well motivated enough, i.e., why do we need to select a subset to perform data augmentation. Although some augmentation techniques are computationally expensive, there are many ``easy'' augmentations (that are also commonly used like random cropping, flipping, permutation, etc.) and these augmentations are not computationally prohibitive.
2. Even if we leave the motivation concern aside, the current experiments do not support the authors' claim. The authors claim that by selecting a subset to perform augmentations, we can speed up the overall training time by several times. Ideally the experiments should compare the training time of (1) train the network using the whole dataset while performing augmentations on all data and (2) traing the network using the whole dataset while only performing augmentations on the selected subset. If the training of (2) is several times faster than (1), then the authors' claim is well supported. However, the current experiments only show that training on a selected subset is much faster than training on the whole dataset. This speedup is a result of fewer training steps and has nothing to do with the data augmentation.

---

> ### Author Response · Authors · 2022-08-02
> **Response to Reviewer xtob**
>
>
> We thank the reviewer for acknowledging the rigor of our theoretical analysis and our extensive empirical evaluations. However, we believe that there is a fundamental misunderstanding of our experimental results. Indeed as we clarify below, all our experiments show the benefit of our method *in presence of standard (weak) data augmentations*, and our results in Table 4, Table 9 and Figure 5 in Appendix D.5 shows the speedup provided by our method in the exact setup as suggested by the reviewer. We also discuss the motivation of our work in general comments above. We hope the reviewer can revisit the score based on our clarifications.
>
> **Motivation**: Please see the general comments above where we discuss the four use cases of our method - (1) When strong augmentation is expensive (2) When augmented examples are appended to the training data (3) When data is larger than the training budget (4) When data contains mislabeled examples.
>
> **Weak augmentation**: As correctly mentioned by the reviewers, default (weak) augmentation is cheap and effective, and we would like to emphasize that **all our experiments are in presence of standard (in-place) augmentations**, where we train on transformed images (with e.g. crop and rotation) of CIFAR-10 instead of original images, at every iteration. Similarly, all the improvements are quantifying the extra benefits of strong augmentation, *on top of* the effect of weak augmentation.
>
> **Speedup of training on whole data while augmenting full data vs subsets**: We respectfully disagree with the reviewer that our method does not result in speedup. Indeed, we addressed the exact same setup as suggested by the reviewer in our experiments (please also see general comments for a list of use cases of our method). In particular, Table 3, 4 cover the reviewers suggested experiments and verify that (b) training on whole data and augmenting subsets is much faster than (a) training on whole data and augmenting the whole data. The corresponding speedup measurements can also be found in Appendix D.5, Table 9, and Figure 5. Indeed, as can be seen in Table 9, the training of (b) is several times faster than (a) - e.g. for the augmentation method of [1] applied to CIFAR10/ResNet,20 it is 3.43x faster to train on the whole dataset and augment our coresets of size 30% than to train and augment the whole dataset. At the same time we achieve 75% of the accuracy improvement of training on and augmenting the full data with [1]. Both cases include standard (weak) augmentation on the full data.
>
> [1] Sen Wu, Hongyang Zhang, Gregory Valiant, and Christopher Ré. On the generalization effects of linear transformations in data augmentation. In International Conference on Machine Learning, pages 10410–10420. PMLR, 2020.

---

> > ### Comment · Reviewer_xtob · 2022-08-04
> > **Response to Authors**
> >
> > Thanks for the authors' response. I think it addressed my concerns about the motivation and the experimental speed-up. The proposed method does provide a reasonable trade-off between the training time and the accuracy when we want to utilize time-consuming augmentations. As a result, I have raised my score.
> >
> > One more additional question: The selection of subsets is formulated as a submodular maximization and solved using the greedy method. How much time does it cost to solve the submodular maximization at each iteration? Is this time cost included when you report the speed-up ratio?

---

> > > ### Author Response · Authors · 2022-08-06
> > > **Response to Reviewer xtob**
> > >
> > > Thank you very much for taking the time to read our clarifications, and updating the review. The time of finding the subsets is indeed included in the speed-up ratio. The *total* wall-clock time for finding the coresets (assuming the selection happens at every epoch) during the entire training can be found in Table 10 of the Appendix (e.g. finding 30% subsets takes only 10.52s on Caltech256, and 8.09s on TinyImageNet). We note that the coresets are extracted from every class separately and the timings reported in Table 10 are based on finding coresets *sequentially* from all classes. In practice, the coresets can be found in parallel from different classes, and since selection only happens every 5-15 epochs, actual speeds are much faster than reported in the table.

---

### Official Review · Reviewer_y9hQ · 2022-07-11

**Rating:** 6
**Confidence:** 3
**Soundness:** 3 good
**Presentation:** 3 good
**Contribution:** 3 good

**Summary:**

The paper first studies the relationship between data augmentation and the singular values of Jacobian in an over-parametrized network. Then, the authors propose a method to select a subset of data augmented training data to speed up the training of the neural networks. Empirically, the proposed approach achieves improved results than the random data argumentation subset selection and speeds up the training on standard image classification tasks.

**Questions:**

Comments:
* As mentioned above, I wonder if there is a useful application for their proposed approach.
* CIFAR-10 baseline (with standard data augmentation) for ResNet20 seems to be lower than the reported values in the literature, where you can above 95% test accuracy.

Minor comments:
* In Equation 3, the eigenvalue $\lambda$ is not defined.
* In Algorithm 1, it would be helpful if variables $V_c$ and $T_i$ were defined.


** After Rebuttal **
I acknowledge that I have read the author's response and other reviews. Thank you for the clarification on the use-cases of the proposed approach. As noted by other reviewers, I believe that including these points more clearly would strengthen the paper.

Regarding the accuracy of the CIFAR-10 baseline, I apologize for the confusion. I thought 95 percent accuracy can be obtained on ResNet-20 on CIFAR-10.

**Limitations:**

I believe that the work does not have any potential negative societal impact.

**Strengths And Weaknesses:**

The topic of the paper is interesting and relevant to the Neurips community. The technical details of the paper look correct, and the authors properly cite important relevant works. The empirical analysis in the paper is detailed, and the authors provide code to reproduce experiments in the paper. It is convincing that the proposed approach can select a subset of augmented data helpful in improving the training accuracy (with reduced dataset) and speed up training.

However, I am still unclear about the core motivation and application of the paper. More specifically, in what cases selecting a subset of the training data is useful (especially when using the full augmentation typically achieves the best test accuracies)? It would be helpful if the authors provided a specific use case of the proposed approach. Moreover, the authors did not fully describe the limitations of their proposed approach. Does the proposed approach work for smaller networks (under-parameterized) or when the dataset size is small? Discussing these limitations in the paper would be helpful to the readers.

---

> ### Author Response · Authors · 2022-08-02
> **Response to Reviewer y9hQ**
>
> Thank you for the review and acknowledging the soundness of our theoretical analysis and detailed empirical evaluations. We hope that our clarifications below can help address the concerns mentioned.
>
> **Core motivation and application of paper**: Please refer to general comments above where we discuss the four use cases of our method - (1) When strong augmentation is expensive (2) When augmented examples are appended to the training data (3) When data is larger than the training budget (4) When data contains mislabeled examples.
>
> **Smaller networks/datasets and limitations**: Our approach is applicable to smaller networks. However, it is mostly useful for large and medium sized datasets on which strong data augmentation or appending augmented examples to training data is expensive. For very small datasets where training is fast one may simply augment and train on all the examples. Besides, on easy datasets such as MNIST where data augmentation does not provide significant gains in performance, our method is naturally not applicable. We will discuss the limitations in the revised version. We also note that our analysis of the effect of data augmentation on singular values and vectors holds even if the network is not overparameterized. Only Lemma 4.2 in the main paper relies on the overparameterization assumption.
>
> **CIFAR-10/ResNet20 accuracy**: Our baseline accuracy for Cifar10/ResNet20 is consistent with the literature. With standard data augmentation, CIFAR-10/ResNet20 achieves 91.25% test accuracy based on the original paper [1]. We based our CIFAR-10/ResNet20 experiments off [this implementation](https://github.com/akamaster/pytorch_resnet_cifar10) which reports 91.73% test accuracy.
>
> **Variable definitions**: Thank you for pointing these out. $V_c$ is the index set for class $c$, and $T_i$ is defined in Sec. 3. We will clarify the definitions better in the revision.
>
> [1] He, Kaiming, et al. "Deep residual learning for image recognition." Proceedings of the IEEE conference on computer vision and pattern recognition. 2016.

---

### Official Review · Reviewer_rqcs · 2022-07-13

**Rating:** 6
**Confidence:** 4
**Soundness:** 3 good
**Presentation:** 4 excellent
**Contribution:** 3 good

**Summary:**

This paper tackles the problem of finding subsets of a data set on which to apply data augmentation, such that the performance is as close as possible to augmenting the whole data set. First, the paper models data augmentation as additive perturbations of the data and provides a theoretical analysis of the effect of perturbations on the singular values of the network Jacobian. Then, it proposes a method to select a subset of the data that aims at matching the alignment of the Jacobian of the fully augmented data with the residual vector. Finally, the paper evaluates the proposed method on several data sets and architectures, comparing with selecting random subsets and max-loss, a previously proposed method.

**Questions:**

Besides the concerns discussed in the previous section, I would like to ask the authors whether they could elaborate on how the theoretical analysis and the conclusions contribute to our understanding of how data augmentation (or additive perturbations) impact overfitting and generalisation. In particular, I refer to Section 4.3, where this conclusions are mentioned. However, I either do not understand completely the relationship between the results and overfitting and generalisation, or the connection is just weak.

I have another comment or suggestion regarding a sentence in the introduction that I find a bit misleading:

"We demonstrate the effectiveness of our approach applied to training ResNet18, ResNet20, ResNet32, ResNet50, Wide-ResNet on CIFAR10, CIFAR10-IB, SVHN, Caltech256, TinyImageNet, and ImageNet compared to random and max-loss baselines"

This gives the impressions that multiple architectures are tested all on multiple data sets, while in fact, only one or maximum two architectures are tested per data sets and the evaluation on some of the data sets, for instance ImageNet, is quite limited.

Finally, a few minor typing errors:

* In Table 1, the heading of the first column is Model/Dataset, but it should read Dataset/Model.
* In page 8, "We use [37] for..." reads weirdly.
* I would use "Table" instead of just "Tab.", which reads more easily and uses just one more character.

**Limitations:**

I have discussed the limitations above. In particular, regarding limitations, I kindly encourage the authors to more explicitly and openly discuss the limitations of the paper. I believe it is a good paper, the results are interesting and relevant, but it is important to discuss, in my opinion, the limitation of modelling data augmentation as additive perturbation, the fact that simple data augmentation methods are effective and computationally cheap and that the empirical results of coresets provide only moderate benefits compared to other methods.

**Strengths And Weaknesses:**


As main strength, I highlight that this paper provides a sound theoretical analysis of how additive perturbations of the inputs impact the eigenspace of the Neural Tangent Kernel (NTK), or the singular values of the model Jacobian. The analysis concludes that input perturbation change proportionally the smaller singular values, while the larger singular values are affected less.

Another strength of the paper is the breadth of the empirical evaluation, in which CIFAR-10, SVHN, Caltech256, Tiny ImageNet and ImageNet are used as benchmarks, though with large differences in the depth of the analysis across data sets.

Finally, regarding strengths, I would like to point out that the paper is clearly written, it is mostly easy to follow and the structure is reasonable.

Regarding weaknesses, my first concern is about the generality of modelling data augmentation as additive perturbations, following the work by Rajput et al. (2019). While I acknowledge that simplifying data augmentation as something that can be easily modelled mathematically is currently necessary, and I do not reject the significance of the results of this paper because of this assumption, I do argue that it is important to also acknowledge the limitation of this modelling assumption. In my opinion, this limitation is barely discussed and, on the contrary, the paper argue that the results can be extended to more complex augmentations, while this is barely analysed in the paper, besides a short section and one figure in the appendix. Data augmentation comprises a range of complex transformations that, in the case of natural images, encode properties of the human visual system. Take, for instance, the case of horizontal flips, which largely preserve the label of the image by applying a transformations that is fundamentally different to additive perturbations. However, the paper makes this strong modelling assumption but then uses the term "data augmentation" to talk about the conclusions from using additive perturbations. As a comparison, Rajput et al. (2019) do acknowledge this limitation and discuss the need to extend the theoretical analyses to practical augmentation methods.

Another concern has to do with the overall motivation for the proposed method, that is for the need to select subsets of data on which to apply data augmentation. On the second sentence of the abstract, the authors write that "the most effective augmentation techniques become computationally prohibitive for even medium-sized datasets". This statement is extended in the introduction, where the authors cite several papers on so-called _automatic data augmentation_. First, these methods are indeed disproportionally expensive in computational terms, while providing only marginal improvements (in the best case) with respect to traditional, simple, cheap data augmentation techniques (Pérez and Wang, 2017). The paper also mentions that performing data augmentation increases "the training time by orders of magnitude". However, it has been shown that the largest benefits of data augmentation are achieved by very simple and cheap transformations such as horizontal flips and translations, and more complex transformations only marginally improve the performance further (Hernández-García and König, 2018). Such simple transformations largely increase the performance without increasing the training time. By way of illustration, the previously cited paper shows that training with 50 % of CIFAR-10 _and_ data augmentation on the full available set achieves more than 95 % of the _full_ accuracy in about half the training time. Therefore, given this result, is it really necessary to develop methods to select subsets given a full data set? In fact, the present paper does not stick to so-called automatic augmentation methods, but also studies CutOut, which is computationally very cheap to perform. Again, if CutOut is considered a strong augmentation method, then what is the need for improving the computational efficiency?

Related to the concerns discussed above, my last main concern is related to the effectiveness of the proposed method given the results and the degree to which the conclusions and claims reflect the results. For example, the conclusion from Table 1 is that "the improvement from augmenting the coresets is significantly larger than random data points". However, a close inspection of the results in the table casts some doubts about the strength of the statement. On CIFAR-10 and ResNet20, the improvement of augmenting the coresets with respect to no augmentation (right-most column) is better than augmenting random sets with 0.1 and 0.2 % subsets. With 0.5 and 1 % subsets, the improvement is about the same. On Caltech256 with ResNet18, the improment on coresets is actually worse than on random sets with 5 and 10 % of the data, and about the same in the rest of the cases. Furthermore, the trend is the opposite than on CIFAR-10, which is confusing. Therefore, I would humbly suggest to adjust the claims regarding this set of results. In particular, these sentences:

* "the augmented coresets outperform augmented random subsets by a large margin": this is not true, as discussed above.
* "the improvement from augmenting the coresets is significantly larger than random data points": this is not true either, by extension.
* "We see that while the non-augmented coreset performs worse than random in most cases": this is also not true, by inspecting the table.

In Table 4, we also see that coresets provide better improvement on average than max-loss, but not consistently in all cases and by a small margin. In particular, the averages across data sets and architectures with 5, 10 and 30 % sets are:

|          | 5 %    | 10 %   | 30 %   |
|----------|--------|--------|--------|
| max-loss | 33.2 % | 45.7 % | 74.8 % |
| coresets | 34.7 % | 49.7 % | 76.6 % |

While coresets indeed provide better improvement on average, the difference is very small, especially if we take into account that that these numbers are the improvement of the performance from no augmentation to full augmentation. The small margin becomes relevant since the speed up of coresets is smaller than that of max-loss, but the authors claim that "[w]hile our method yields similar or slightly lower speed-up to the max-loss policy and random approach respectively, our resulting accuracy outperforms these two approaches". I would kindly suggest to adjust this claim too.

References

* Pérez and Wang. [The effectiveness of data augmentation in image classification using deep learning](https://arxiv.org/abs/1712.04621). 2017.
* Hernández-García and König. [Data augmentation instead of explicit regularization](https://arxiv.org/abs/1806.03852). 2018

---

> ### Author Response · Authors · 2022-08-02
> **Response to Reviewer rqcs (2/2)**
>
> **Improvement over max loss (Table 4)**: We believe that there is a misunderstanding of our results, and highlight that all our experiments are indeed *in the presence of standard (weak) augmentation*. Hence, the smaller improvements are expected and are not negligible. All the SOTA augmentation strategies, including [1,3,4,5] achieve accuracy improvement of up to 2% for CIFAR-10 experiments. Thus, the smaller percentage improvement of our coreset augmentation is indeed considerable.
> We will revise the following sentence to reflect that “on average” we achieve improvement over max-loss: "[w]hile our method yields similar or slightly lower speed-up to the max-loss policy and random approach respectively, our resulting accuracy outperforms these two approaches on average"
>
> **Clarification on theoretical analysis in Sec 4.3**: Recent studies have revealed that the Jacobian matrix of common neural networks is low rank. That is there are a number of large singular values and the rest of the singular values are small. Based on this, the Jacobian spectrum can be divided into information and nuisance spaces. Information space is a lower dimensional space associated with the prominent singular value/vectors of the Jacobian. Nuisance space is a high dimensional space corresponding to smaller singular value/vectors of the Jacobian. While learning over information space is fast and generalizes well, learning over nuisance space is slow and results in overfitting [6]. Importantly, recent theoretical studies connected the generalization performance to small singular values (of the information space) [7]. Our results show that additive perturbations relatively enlarge the smaller singular values of the Jacobian in a *stochastic* way. This benefits generalization in 2 ways. First, this stochastic behavior prevents overfitting along any particular singular direction *in the nuisance space*, as stochastic perturbation of the *smallest* singular values results in a stochastic noise to be added to the gradient at every training iteration. This prevents overfitting (thus a larger training loss as shown in Appendix D), and improves generalization [8,9]. Second, additive perturbations improve the generalization by enlarging the smaller (useful) singular values that lie in the *information space* (Lemma 4.2), and speeding up learning along them.
>
> **Revising final sentence of intro**: Thank you for pointing these out, we updated the manuscript to better clarify this. We are also running more experiments on ImageNet to add to the final version of the manuscript.
>
> [1] Ekin D Cubuk, Barret Zoph, Dandelion Mane, Vijay Vasudevan, and Quoc V Le. Autoaugment: Learning augmentation strategies from data. In Proceedings of the IEEE/CVF Conference on Computer Vision and Pattern Recognition, pages 113–123, 2019.
>
> [2] Calvin Luo, Hossein Mobahi, and Samy Bengio. Data augmentation via structured adversarial perturbations. arXiv preprint arXiv:2011.03010, 2020.
>
> [3] Terrance DeVries and Graham W Taylor. Improved regularization of convolutional neural networks with cutout. arXiv preprint arXiv:1708.04552, 2017.
>
> [4] Dan Hendrycks, Norman Mu, Ekin D Cubuk, Barret Zoph, Justin Gilmer, and Balaji Lakshminarayanan. Augmix: A simple data processing method to improve robustness and uncertainty. arXiv preprint arXiv:1912.02781, 2019.
>
> [5] Sen Wu, Hongyang Zhang, Gregory Valiant, and Christopher Ré. On the generalization effects of linear transformations in data augmentation. In International Conference on Machine Learning, pages 10410–10420. PMLR, 2020.
>
> [6] Oymak, Samet, Zalan Fabian, Mingchen Li, and Mahdi Soltanolkotabi. Generalization guarantees for neural networks via harnessing the low-rank structure of the jacobian. arXiv preprint arXiv:1906.05392 (2019).
>
> [7] Sanjeev Arora, Simon Du, Wei Hu, Zhiyuan Li, and Ruosong Wang. Fine-grained analysis of optimization and generalization for overparameterized two-layer neural networks. In International Conference on Machine Learning, pages 322–332. PMLR, 2019
>
> [8] Zhu, Zhanxing, Jingfeng Wu, Bing Yu, Lei Wu, and Jinwen Ma. "The anisotropic noise in stochastic gradient descent: Its behavior of escaping from sharp minima and regularization effects." arXiv preprint arXiv:1803.00195 (2018).
>
> [9] Jastrzębski, Stanisław, Zachary Kenton, Devansh Arpit, Nicolas Ballas, Asja Fischer, Yoshua Bengio, and Amos Storkey. "Three factors influencing minima in sgd." arXiv preprint arXiv:1711.04623 (2017).

---

> > ### Comment · Reviewer_rqcs · 2022-08-04
> > **Response to authors 1**
> >
> > I thank the authors for the clarifications for continuing the discussion.
> >
> > ### Limitations of the additive perturbation model:
> >
> > I appreciate the acknowledgement of the limitations of the additive perturbation model and the extended discussion about when it is a reasonable approximation and when it is not. As mentioned in my review, I believe the original submission is not transparent enough about the limitations, therefore I think the paper would be more solid if the authors indeed incorporate a discussion along the lines of the response here.
> >
> > ### Motivation of our work
> >
> > The explanation of the scenarios where the proposed method is of use is very clear and insightful. Again, I kindly suggest to incorporate this in the manuscript.
> >
> > ### Improvement over random subset augmentation (Table 1)
> >
> > > we emphasize that all our experiments, including Table 1, are in the presence of standard (weak) augmentation
> >
> > Should I understand that even the models corresponding to the columns "No Aug." were trained with "standard (weak) augmentation"? If so, I strongly suggest to use a less confusing wording.
> >
> > > the way improvement is calculated in Table 1 [...] is not a good way to measure improvement of augmentation over no-aug
> >
> > I actually agree that the improvement with respect not augmenting (weakly augmenting, actually, it seems) is not what you are interested in, since you want to quantify the effectiveness of the coresets _to be augmented_. In fact, a metric that would reflect this and seems to speak in favour of the proposed method is the relative improvement with respect to augmenting random sets, that is $$\frac{acc_{coreset} - acc_{random}}{acc_{random}}$$ where the accuracy is on both cases that by the augmented sets.
> >
> > Regarding the updated claims, I agree that the new versions more faithfully reflect the results.
> >
> > ### Improvement over max loss (Table 4):
> >
> > I had indeed understood that were "No aug." is written, the models were trained without _any_ data augmentation. I would make this clearer in the manuscript. I still think that the improvement with respect to max-loss is small, but it is true, as I originally acknowledged, that it is indeed an improvement.
> >
> > ### Clarification on theoretical analysis in Sec 4.3:
> >
> > Thanks for the explanation!

---

> > > ### Author Response · Authors · 2022-08-06
> > > **Manuscript Updated**
> > >
> > > We sincerely thank the reviewer for the very helpful feedback. We appreciate all the great suggestions that helped improve our manuscript.
> > > We have made the following changes based on your suggestions in the updated manuscript (underlined in blue)
> > >
> > > **Limitations of the additive perturbation model**: We added the discussion to Sec 3 and conclusion.
> > >
> > > **Motivation**: we clarified the motivations and use cases in the introduction (examples are also discussed in the related work).
> > >
> > > **Improvement over random subset augmentation (Table 1)**: We thank the reviewer for the suggestion on how to calculate the improvements. We agree that this is a much better way to calculate the improvement of augmenting coresets vs random subsets than what we had previously. We dropped the improvement column from the table and the better performance of the strongly augmented coresets over random subsets can now be observed directly from the accuracy values in the table. We realized that for the suggested improvement metric to clearly show a trend, we need to repeat some of our experiments a few more times to make the standard deviations of the accuracies consistent. We will update the table with the improvement if the experiments finish by the end of the discussion period. Otherwise, we will add the scores to the final version of the manuscript.
> > >
> > > **Improvement over max loss (Table 4)**: We thank the reviewer for bringing this to our attention. We updated Table 4 accordingly. Thank you!

---

> ### Author Response · Authors · 2022-08-02
> **Response to Reviewer rqcs (1/2)**
>
> We appreciate the comprehensive review and acknowledging the soundness of our analysis and breath of our empirical evaluations. We would like to address the following points raised:
>
> **Limitations of the additive perturbation model**: We do agree and acknowledge that additive perturbations are indeed limited in modeling all real-world data augmentation techniques. We note that while the additive perturbation model is unable to effectively capture augmentations like rotations and crops in general (i.e. without constraints), most real-world augmentations are bounded to preserve the regularities of natural images (e.g. AutoAugment finds that a 6 degree rotation is optimal for CIFAR10). As such, under local smoothness of images, our additive perturbation still can be applied to model many bounded transformations such as small rotations, crops, shearing etc. We believe this is the reason we see the effects of additive augmentation on the singular spectrum holds even under real-world augmentation settings (Fig. 3, Appendix) such as rotations, crops, and stronger augmentations like AutoAugment.
>
> We acknowledge, however, that our additive perturbation model is indeed limited when applied to augmentations that cannot be reduced to perturbations, such as horizontal/vertical flips and large translations.  We tried to partially address this by extending our theoretical analysis to augmentations modeled as arbitrary linear transforms (e.g. as mentioned, horizontal flips) in Sec. B.5. in Appendix. We also empirically showed the effectiveness of our subset selection technique under more complex data augmentations. However, theoretical analysis of complex data augmentations is indeed an interesting direction for future work.
>
> We will incorporate the feedback to make the limitations more explicit by clarifying that additive perturbation model is theoretically limited only to small translations, crops, rotations under local smoothness constraint, and for other pixel-wise augmentation methods such as sharpening, blurring, and color distortions [1], where $\epsilon_0$ is small. This does also include state-of-the-art augmentation techniques such as structured adversarial perturbation [2], in which pixel intensities are changed minimally. However, other augmentations like large translations, crops, flips etc. are better modeled using the linear transformation model (Sec. B.5., Appendix).
>
> **Motivation of our work**: please see the general comments above where we list four use cases for our method - (1) When strong augmentation is expensive (2) When augmented examples are appended to the training data (3) When data is larger than the training budget (4) When data contains mislabeled examples.
>
> **Improvement over random subset augmentation (Table 1)**: We thank the reviewer for bringing Table 1 to our attention. First, we emphasize that all our experiments, including Table 1, are in the presence of standard (weak) augmentation. Thinking more about the reviewers feedback, we believe that the way improvement is calculated in Table 1, i.e. $\frac{acc_{aug} - acc_{noaug}}{acc_{noaug}}$, is not a good way to measure improvement of augmentation over no-aug (weak augmentation). This metric is indeed sensitive to the $acc_{noaug}$, which explains the discrepancies in results pointed out by the reviewer. To address this, we dropped the improvement columns from Table 1, and instead we compare the effectiveness of training on and augmenting coresets over random subsets based on Table 6. In Table 6, we compared the performance of training on random subsets while augmenting coresets vs (other) random subsets. As the base accuracy of training on random subsets with weak (standard) augmentation is the same in both cases, we can correctly quantify the improvement of augmenting coresets over random subsets. Comparing the final accuracies, see that augmenting coresets are indeed much more effective than augmenting random subsets. We believe this fairer comparison satisfies the following claims (we dropped ‘significantly’ from the paper based on reviewer’s suggestion, but we get improvement in all cases in the following cases):
>
> > "the augmented coresets outperform augmented random subsets."
>
> > "the improvement from augmenting the coresets is larger than random data points."
>
> We also drop the following sentence suggested by the reviewer, as it is not necessary for our argument to hold.
>
> > "We see that while the non-augmented coreset performs worse than random in most cases".
>
> We will also switch Sec 6.1 and 6.3, to discuss the improvement of coreset augmentation when added to (1) random subsets, (2) full data, and (3) coresets vs random subsets.

---

### Author Response · Authors · 2022-08-02
**General comment: Motivation and experimental setup**

We thank all the reviewers for their valuable feedback and acknowledging the soundness of our analysis and breath of our empirical evaluations. Here, we clarify our experimental setup and motivation, and list the scenarios where our method improves efficiency and robustness of training.

### **Weak Augmentation**

First, as correctly mentioned by the reviewers, standard (weak) augmentation is cheap and effective, and we would like to emphasize that **all our experiments are in presence of standard (in-place) augmentations**, where we train on transformed images (with e.g. crop and rotation) of CIFAR-10 instead of original images, at every iteration. Similarly, all the improvements are quantifying the extra benefits of strong augmentation, **on top of** the effect of weak augmentation.

### **Motivation and use cases**
Second, we clarify that our method boosts speed and robustness of training in the following four scenarios:

1. **When strong augmentation is expensive**: Strong augmentation is often expensive but is crucial to get SOTA accuracy [1,2,3,4]. For example, [2] increases training time by 2.8x on Caltech256/ResNet18. Our method introduces an efficient way to only strongly augment a subset of data to achieve near SOTA accuracy with much smaller computational cost.

2. **When augmented examples are appended to the training data**: As Table 8 in Appendix D.4 shows, **appending transformed examples** to the training data is often much more effective in boosting the test accuracy than training only on the transformed examples (strong or weak in-place augmentation). In fact, in-place augmentation can even hurt the performance, e.g. on Caltech256. Importantly, appending augmentation is often much more effective than training on two transformed examples in place of every original training data (In-place 2x), which has the same computational cost as that of training on original images appended with only one transformation per image. This is the motivation for several (cheap or expensive) SOTA augmentation methods that add the augmented examples to the training data. Here, even if producing transformations are cheap (e.g. CutOut [5], AugMix [6], AutoAugment [7]), such methods significantly increase the size of the training data (e.g., AugMix increases the effective dataset size to 3x, by adding 2 transformed examples per image). More expensive methods such as [4] (used in Tables 1,4,5) increases training time of ResNet20 on CIFAR10 by 13x on an Nvidia A40 GPU when used on full data. Our method can effectively reduce training time by selecting a small subset to be transformed and appended to the training data to boost the performance. Tables 3, 4 show the accuracy improvement and Table 9 in Appendix D.5 shows the speedups of training on full data while augmenting our coresets. For example we see that for the augmentation method of [4] applied to CIFAR10/ResNet20 it is 3.43x faster to train on the whole dataset and augment our coresets of size 30% than to train and augment the whole dataset. At the same time we achieve 75% of the accuracy improvement of training on and augmenting the full data with [4]. Both cases include standard (weak) augmentation on the full data.

3. **When data is larger than the training budget**: In this case, our experiments show that one can train on random subsets (with standard in-place augmentation) and augment coresets (by strong augmentation and/or appending transformations) to achieve a superior performance. In Table 6, we show that we can achieve 71.99% test accuracy on ResNet50/ImageNet when training on and augmenting only 30% subsets for 90 epochs. For example, compared to AutoAugment [7] despite using only 30% subsets, we achieve 86% of the original reported accuracy while boasting 5x speed-up in training time compared to their original training. The experiments involve default(weak) augmentation on the subsets.

4. **When data contains mislabeled examples**: In this situation, augmentation methods that append transformations to the data (e.g. by max loss) degrade the performance by selecting and appending several noisy labels. However, augmenting coresets achieve a superior accuracy. In Table 5, we show that training on and strongly augmenting 50% subsets using our method on CIFAR-10 with 50% noisy labels achieves 76.20% test accuracy. This actually improves performance compared to training on and strongly augmenting the full data (75.87% test accuracy). All numbers include default (weak) augmentation on full data.

---

> ### Author Response · Authors · 2022-08-02
> **References**
>
> [1] Christopher Bowles, Roger Gunn, Alexander Hammers, and Daniel Rueckert. Gansfer learning: Combining labelled and unlabelled data for gan based data augmentation. arXiv preprint arXiv:1811.10669, 2018.
>
> [2] Philip TG Jackson, Amir Atapour Abarghouei, Stephen Bonner, Toby P Breckon, and Boguslaw Obara. Style augmentation: data augmentation via style randomization. In CVPR Workshops, volume 6, pages 10–11, 2019.
>
> [3] Joseph Lemley, Shabab Bazrafkan, and Peter Corcoran. Smart augmentation learning an optimal data augmentation strategy. Ieee Access, 5:5858–5869, 2017.
>
> [4] Sen Wu, Hongyang Zhang, Gregory Valiant, and Christopher Ré. On the generalization effects of linear transformations in data augmentation. In International Conference on Machine Learning, pages 10410–10420. PMLR, 2020.
>
> [5] Terrance DeVries and Graham W Taylor. Improved regularization of convolutional neural networks with cutout. arXiv preprint arXiv:1708.04552, 2017.
>
> [6] Dan Hendrycks, Norman Mu, Ekin D Cubuk, Barret Zoph, Justin Gilmer, and Balaji Lakshminarayanan. Augmix: A simple data processing method to improve robustness and uncertainty. arXiv preprint arXiv:1912.02781, 2019.
>
> [7] Ekin D Cubuk, Barret Zoph, Dandelion Mane, Vijay Vasudevan, and Quoc V Le. Autoaugment: Learning augmentation strategies from data. In Proceedings of the IEEE/CVF Conference on Computer Vision and Pattern Recognition, pages 113–123, 2019.

---

> ### Comment · Reviewer_rqcs · 2022-08-04
> **Good summary and clarification. Consider incorporating into the manuscript.**
>
> I thank the authors for the clarification about the experimental setup regarding baseline weak augmentations and the comprehensive explanation of the scenarios where coresets may be of use. I think the experimental setup about the baseline models including weak augmentations was not clearly explained in the paper (in fact, that the wording "No aug." was used is confusing). Therefore, I would suggest to improve the clarity of this aspect. Similarly, in my opinion, the paper would be stronger by including such a discussion about the scenarios where the proposal may be beneficial.

---

> > ### Author Response · Authors · 2022-08-06
> > **RE: Incorporating into the manuscript**
> >
> > Many thanks for the valuable feedback and suggestions to further improve the clarity of the manuscript. In our revised version, we have clarified the use cases (please see the updated introduction), and that weak augmentation is used in all the experiments (please see the updated experiments section).

---

> > > ### Comment · Area_Chair_fcxR · 2022-08-08
> > > **page limit**
> > >
> > > Dear authors,
> > > Thanks for updating your paper. Please make sure to respect the page limit (<=9 pages, no limit for appendices). -AC

---

> > > > ### Author Response · Authors · 2022-08-09
> > > > **Manuscript updated to fit page limit**
> > > >
> > > > Thank you for the comment. We were under the impression that we can have an extra page to incorporate the reviewers’ feedback. We moved some materials to the appendix to fit within the page limit (9 pages) based on your comment. Thank you!

---

### Meta-Review · Area_Chair_fcxR · 2022-08-26

**Recommendation:** Accept
**Confidence:** Less certain

**Metareview:**

This work demonstrates that it can be sufficient to apply data-augmentation only on a core-set of the data to achieve accuracy comparable to augmenting the full dataset. These findings are supported by theoretical arguments in the NTK framework, and by empirical evaluation. The proposed method can provide a trade-off between the training time and the accuracy when data-augmentation is costly.

The reviewers noted that the restriction to additive perturbations might be a limitation of the proposed approach and suggested to incorporate a discussion of these limitations and possible use cases in the final version of the paper (such as also promised in the rebuttal).

**Award:**

No

---

### Decision · Program_Chairs · 2022-09-14

Accept